# ITERATED GRAPH NEURAL NETWORK SYSTEM

## ABSTRACT

We present Iterated Graph Neural Network System (IGNNS), a new framework of Graph Neural Networks (GNNs), which can deal with undirected graph and directed graph in a unified way. The core component of IGNNS is the Iterated Function System (IFS), which is an important research field in fractal geometry. The key idea of IGNNS is to use a pair of affine transformations to characterize the process of message passing between graph nodes and assign an adjoint probability vector to them to form an IFS layer with probability. After embedding in the latent space, the node features are sent to IFS layer for iterating, and then obtain the high-level representation of graph nodes. We also analyze the geometric properties of IGNNS from the perspective of dynamical system. We prove that if the IFS induced by IGNNS is contractive, then the fractal representation of graph nodes converges to the fractal set of IFS in Hausdorff distance and the ergodic representation of that converges to a constant matrix in Frobenius norm. We have carried out a series of semi-supervised node classification experiments on citation network datasets such as citeser, Cora and PubMed. The experimental results show that the performance of our method is obviously better than the related methods.

## 1 INTRODUCTION

GNN (Scarselli et al., 2009) has been proved to be effective in processing graph structured data, and has been widely used in natural language processing, computer vision, data mining, social network and biochemistry. In recent years, GNN has developed a variety of architectures, such as GCN (Kipf & Welling, 2017), GraphSAGE (Hamilton et al., 2017), GAT (Veličković et al., 2018), DGI (Veličković et al., 2019), GIN (Xu et al., 2019), GCNII (Ming Chen et al., 2020) and GEN (Li et al., 2020). These architectures have a common feature, that is, the representation of each node is updated using messages from its neighbors but without distinguishing the direction (or angle) of message passing between two nodes. Recent studies have shown that considering directed message passing between nodes can improve the performance of GNN and achieve success in related fields. For example, DimeNet (Klicpera et al., 2020) considers the spatial direction from one atom to another and can learn both molecular properties and atomic forces. R-GCN (Schlichtkrull et al., 2018) and Bi-GCN (Marcheggiani & Titov, 2017; Fu et al., 2019) are models for directed graph, applied in the field of natural language processing. We note that the above direction based model does not consider the bidirectional mixed passing of messages.

But in real life, message passing is interactive in different directions. For example, node A obtains a message from node B. After processing the message, node A not only passes it to the next node C, but also feeds back to node B. Suppose there are only two directions for message passing, forward and backward, represented by 0 or 1, respectively. The symbol space of the first generation message passing path is $\{0, 1\} = \{0, 1\}^1$, and that of the second generation message passing path is $\{00, 01, 10, 11\} = \{0, 1\}^2$. Generally, the symbol space of the $n$-th generation message passing path is $\{0, 1\}^n$ and the size of the symbol space is $2^n$. This means that the scope of message passing spreads with exponent 2. However, in Bi-GCN (similar to Bi-LSTM) and R-GCN architectures, the symbol space is $\{\{0\}^n, \{1\}^n\}$, and its size is 2, which indicates that a lot of information will be lost in the process of message passing (see Appendix A).

How to characterize the above message passing patterns? We use two mappings to represent message passing process in two directions. Then the interactive passing of messages in different directions is equivalent to the composite operation of corresponding mappings. In addition, the direction of

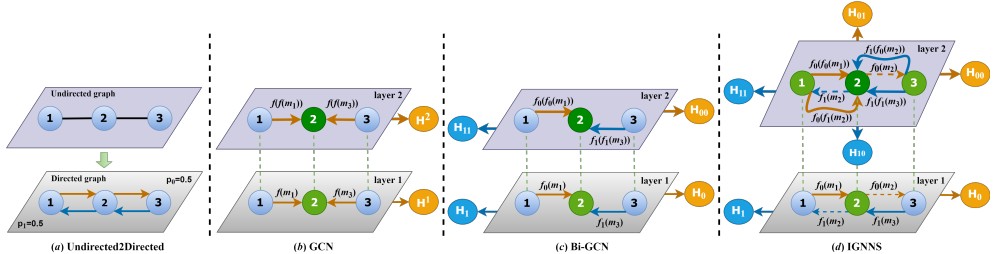

Figure 1: Message passing patterns. Where the symbol $\boldsymbol{H}$ is the representations of all the notes. (a) An undirected graph is transformed into a directed graph in a natural way. (b) Regardless of direction, simply gather information from neighbors. (c) Message is passed in the same direction (forward or backward), and get two hidden representations independently. (d) Message passing not only occurs in the same direction, but also occurs interactively in different directions, which is more in line with the actual situation. For example, in layer 1, node 2 passes the processed message $f_1(m_2)$ to node 1, and then, in layer2, node 1 processes the received message $f_1(m_2)$ and returns the processed message $f_0(f_1(m_2))$ to node 2.

message passing is often random, so we endow the two mappings with an adjoint probability vector to reflect the randomness. Because the symbol space of the iterative path of the Iterated Function System (IFS) with two mappings is also $\{0, 1\}^n$ and the mapping is selected with a certain probability, the iterative process of IFS is similar to the message passing process. In other words, the above message passing pattern can be described perfectly by IFS with probabilities. We naturally present the Iterative Graph Neural Network System (IGNNS), whose core layer is constructed by IFS. Figure 1 describes the differences in message passing patterns among GCN, Bi-GCN and IGNNS. At the same time, we regard undirected graph as a directed graph with equal probability of bidirectional message passing (see Figure 1(a)), so the IGNNS architecture can handle directed graph and undirected graph in a unified way.

## 2 PRELIMINARIES

A graph $\mathcal{G} = (V, E)$ is defined by its note set $V = \{v_1, v_2, ..., v_N\}$ and edge set $E = \{(v_i, v_j)|v_i, v_j \in V\}$. Let $\boldsymbol{A} \in \mathbb{R}^N$ denote the adjacency matrix of $\mathcal{G}$, providing with relational information between nodes. $\boldsymbol{A}[i, j]$ denote $i, j$th element of $\boldsymbol{A}$, $\boldsymbol{A}[i, :]$ means the $i$th row, and $\boldsymbol{A}[:, j]$ means the $j$th column. In this paper, we assume that all nodes of $\mathcal{G}$ are self adjacent, that is $\boldsymbol{A}[i, i] = 1, i = 1, 2, ..., N$. let $\boldsymbol{D} = diag(d_1, d_2, ..., d_N)$ be the degree matrix of $\boldsymbol{A}$, where $d_i = \sum_{j=1}^{N} \boldsymbol{A}[i, j]$.

**Neighborhood Normalization.** There are two ways to normalize $\boldsymbol{A}$. One approach is the following mean-pooling employed by Hamilton et al. (2017) and Veličković et al. (2019) for inductive learning:

$$\boldsymbol{A}_{mp} = \boldsymbol{D}^{-1}\boldsymbol{A}.$$

Another approach is the following symmetric normalization employed by Kipf & Welling (2017):

$$\boldsymbol{A}_{sym} = \boldsymbol{D}^{-\frac{1}{2}}\boldsymbol{A}\boldsymbol{D}^{-\frac{1}{2}}.$$

**Iterated Function System.** A mapping $f : \mathbb{R}^N \to \mathbb{R}^N$ is said to be a contractive mapping on $\mathbb{R}^N$ if there exists a constant $0 < c < 1$ such that $\|f(x_1) - f(x_2)\|_2 < c\|x_1 - x_2\|_2$ for all $x_1, x_2 \in \mathbb{R}^N$. An iterated function system (Hutchinson, 1981) is defined by

$$\text{IFS} = \{\mathbb{R}^N; f_1, f_2, ..., f_m; \boldsymbol{p}\},$$

where each $f_i : \mathbb{R}^N \to \mathbb{R}^N$ is a contractive mapping and $\boldsymbol{p} = (p_1, p_2, ..., p_m)$ is an adjoint probability vector meaning that $f_i$ is selected by probability $p_i$ for each iteration. Hutchinson (1981) showed that there exists a unique nonempty compact set $\mathbb{F}$ such that

$$\mathbb{F} = \bigcup_{i=1}^{m} f_i(\mathbb{F}).$$

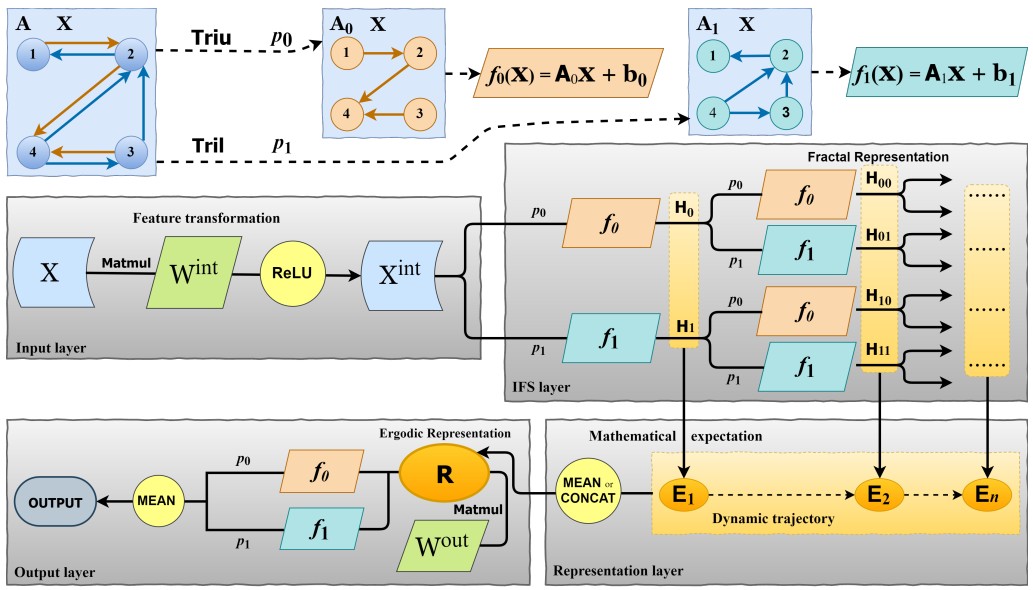

Figure 2: An overview of IGNNS. The upper part of the Figure describes how to generate two affine transformations on $\mathbb{R}^4$, where we use the mean-pooling method to normalize $\boldsymbol{A}$, $p_0 = 0.6$, $p_1 = 0.4$,

$$\boldsymbol{A} = \begin{pmatrix} 1 & 1 & 0 & 0 \\ 1 & 1 & 0 & 1 \\ 0 & 1 & 1 & 1 \\ 0 & 1 & 1 & 1 \end{pmatrix}, \quad \boldsymbol{A}_0 = \begin{pmatrix} \frac{1}{2} & \frac{1}{2} & 0 & 0 \\ 0 & \frac{1}{2} & 0 & \frac{1}{2} \\ 0 & 0 & \frac{1}{2} & \frac{1}{2} \\ 0 & 0 & 0 & 1 \end{pmatrix} \text{ and } \boldsymbol{A}_1 = \begin{pmatrix} 1 & 0 & 0 & 0 \\ \frac{1}{2} & \frac{1}{2} & 0 & 0 \\ 0 & \frac{1}{2} & \frac{1}{2} & 0 \\ 0 & \frac{1}{3} & \frac{1}{3} & \frac{1}{3} \end{pmatrix}.$$

We call $\mathbb{F}$ the fractal set or invariant set of IFS. More conclusions on IFS can be found in the Appendix D. It is well known that there exists a unique probability measure $\mu$ with support $\mathbb{F}$ satisfying the equation

$$\mu = \sum_{i=1}^{m} p_i \mu \circ f_i^{-1}. \tag{1}$$

The probability measure $\mu$ in (1) is called the self-similar measure of IFS with probability vector $\boldsymbol{p}$.

## 3 IGNNS ARCHITECTURE

In this section, we will introduce the architecture of the IGNNS according to the input layer, IFS layer, representation layer and output layer, which is described in Figure 2.

### 3.1 INPUT LAYER

Given a graph structure data $\boldsymbol{X} \in \mathbb{R}^{N \times F}$ of $\mathcal{G} = (V, E)$, called as the feature matrix of node set $V$. A row of $\boldsymbol{X}$ represents the $F$-dimensional feature vector of a node in $V$. Let $\boldsymbol{W}^{\text{int}} \in \mathbb{R}^{F \times H}$ be a learnable parameter matrix, where $H$ is the dimension of the latent space. Then $\boldsymbol{X}\boldsymbol{W}^{\text{int}} \in \mathbb{R}^{N \times H}$. The output of input layer is defined by

$$\boldsymbol{X}^{\text{int}} = \sigma(\boldsymbol{X}\boldsymbol{W}^{\text{int}}) \in \mathbb{R}^{N \times H}, \tag{2}$$

where $\sigma(\cdot)$ is the activation function. Generally, $\text{ReLU}(x) = \max(0, x)$ is used as the nonlinear activation function. Here, each column of $\boldsymbol{X}^{\text{int}}$ is regarded as a point in $\mathbb{R}^N$, so $\boldsymbol{X}^{\text{int}}$ is the set of $H$ points in $\mathbb{R}^N$ and arranged in a certain order. The vector composed of the $i$th component of these points (the $i$th row of $\boldsymbol{X}^{\text{int}}$) is a feature representation of the $i$th node of graph $\mathcal{G}$.

## 3.2 IFS LAYER

Let $\boldsymbol{A}$ be the adjacency matrix of $\mathcal{G}$. Let $triu(\boldsymbol{A})$ denote the upper triangular matrix of $\boldsymbol{A}$ and $tril(\boldsymbol{A})$ denote the lower triangular matrix of $\boldsymbol{A}$. The symmetric normalization of $triu(\boldsymbol{A})$ and $tril(\boldsymbol{A})$ are

$$\boldsymbol{A}_0 = \boldsymbol{D}_0^{-\frac{1}{2}} triu(\boldsymbol{A}) \boldsymbol{D}_0^{-\frac{1}{2}} \text{ and } \boldsymbol{A}_1 = \boldsymbol{D}_1^{-\frac{1}{2}} tril(\boldsymbol{A}) \boldsymbol{D}_1^{-\frac{1}{2}},$$

where $\boldsymbol{D}_0$ and $\boldsymbol{D}_1$ are degree matrices of $triu(\boldsymbol{A})$ and $tril(\boldsymbol{A})$ respectively. Sometimes, we use the mean-pooling of $triu(\boldsymbol{A})$ and $tril(\boldsymbol{A})$, i.e. $\boldsymbol{A}_0 = \boldsymbol{D}_0^{-1} triu(\boldsymbol{A}), \boldsymbol{A}_1 = \boldsymbol{D}_1^{-1} tril(\boldsymbol{A})$. Let $f_0, f_1$ be the two affine transformations on $\mathbb{R}^N$, induced by $\boldsymbol{A}_0, \boldsymbol{A}_1$ respectively, defined as follows:

$$f_0 : x \to \boldsymbol{A}_0 x + b_0, x \in \mathbb{R}^N, b_0 \in \mathbb{R}, f_1 : x \to \boldsymbol{A}_1 x + b_1, x \in \mathbb{R}^N, b_1 \in \mathbb{R},$$

where $b_0$ and $b_1$ are learnable biases, namely add constants $b_0$ and $b_1$ to each component of $\boldsymbol{A}_0 x$ and $\boldsymbol{A}_1 x$ respectively. Constructing iterated function system

$$\text{IFS} = \{\mathbb{R}^N; f_0, f_1; \boldsymbol{p}\},$$

where $\boldsymbol{p} = (p_0, p_1)$ is a learnable adjoint probability vector, satisfying $p_0 > 0, p_1 > 0$ and $p_0 + p_1 = 1$. Using the symbol space $\Omega_m = \{0, 1\}^m$, then for each $\mathbf{i} = (i_1, i_2, ..., i_m) \in \Omega_m$ the length of $\mathbf{i}$ is $m$, denoted as $|\mathbf{i}| = m$, and defining $\boldsymbol{p}_\mathbf{i} = p_{i_1} p_{i_2} \cdots p_{i_m}$ and $f_\mathbf{i} = f_{i_1} \circ f_{i_2} \circ \cdots \circ f_{i_m}$. Let $n$ be the number of iterations of IFS. For IGNNS, $n$ is a preset parameter. The iterative process of IFS is described as follows:

**The first iteration ($|\mathbf{i}| = 1$).** The result of the first iteration is denoted by

$$\mathbb{H}^{(1)} = \{f_0(\boldsymbol{X}^{\text{int}}), f_1(\boldsymbol{X}^{\text{int}})\} = \{\boldsymbol{H}_\mathbf{i}\}_{|\mathbf{i}|=1},$$

where $\boldsymbol{H}_\mathbf{i} = f_\mathbf{i}(\boldsymbol{X}^{\text{int}}), \forall \mathbf{i} \in \Omega_1$. Since IFS selects the iteration branch $f_i$ with probability $p_i$, the mathematical expectation of $\mathbb{H}^{(1)}$ is computed by

$$\boldsymbol{E}_1 = p_0 f_0(\boldsymbol{X}^{\text{int}}) + p_1 f_1(\boldsymbol{X}^{\text{int}}) = p_0 \boldsymbol{H}_0 + p_1 \boldsymbol{H}_1 = \sum_{|\mathbf{i}|=1} \boldsymbol{p}_\mathbf{i} \boldsymbol{H}_\mathbf{i}.$$

If choose to use bias in iterations, then $\boldsymbol{H}_0 = \boldsymbol{A}_0 \boldsymbol{X}^{\text{int}} + \boldsymbol{b}_0, \boldsymbol{H}_1 = \boldsymbol{A}_1 \boldsymbol{X}^{\text{int}} + \boldsymbol{b}_1$, where $\boldsymbol{b}_0$ and $\boldsymbol{b}_1$ are learnable $H$-dimensional vectors.

**The second iteration ($|\mathbf{i}| = 2$).** Using the results of the first iteration as the input of the second iteration, then the result of the second iteration is denoted by

$$\begin{aligned}
\mathbb{H}^{(2)} &= \{f_0(f_0(\boldsymbol{X}^{\text{int}})), f_0(f_1(\boldsymbol{X}^{\text{int}})), f_1(f_0(\boldsymbol{X}^{\text{int}})), f_1(f_1(\boldsymbol{X}^{\text{int}}))\} \\
&= \{f_{00}(\boldsymbol{X}^{\text{int}}), f_{01}(\boldsymbol{X}^{\text{int}}), f_{10}(\boldsymbol{X}^{\text{int}}), f_{11}(\boldsymbol{X}^{\text{int}})\} = \{\boldsymbol{H}_\mathbf{i}\}_{|\mathbf{i}|=2},
\end{aligned}$$

where $\boldsymbol{H}_\mathbf{i} = f_\mathbf{i}(\boldsymbol{X}^{\text{int}}), \forall \mathbf{i} \in \Omega_2$. Note that IFS selects the iteration path $f_\mathbf{i}$ with probability $\boldsymbol{p}_\mathbf{i}$, then the mathematical expectation of $\mathbb{H}^{(2)}$ is computed by

$$\boldsymbol{E}_2 = \sum_{|\mathbf{i}|=2} \boldsymbol{p}_\mathbf{i} f_\mathbf{i}(\boldsymbol{X}^{\text{int}}) = \sum_{|\mathbf{i}|=2} \boldsymbol{p}_\mathbf{i} \boldsymbol{H}_\mathbf{i}.$$

We expand the expression of $\boldsymbol{E}_2$ and perceive its powerful feature representation ability. First,

$$\boldsymbol{H}_{00} = f_0(\boldsymbol{H}_0) = \boldsymbol{A}_0(\boldsymbol{A}_0 \boldsymbol{X}^{\text{int}} + \boldsymbol{b}_0) + \boldsymbol{b}_0, \boldsymbol{H}_{01} = f_0(\boldsymbol{H}_1) = \boldsymbol{A}_0(\boldsymbol{A}_1 \boldsymbol{X}^{\text{int}} + \boldsymbol{b}_1) + \boldsymbol{b}_0,$$

$$\boldsymbol{H}_{10} = f_1(\boldsymbol{H}_0) = \boldsymbol{A}_1(\boldsymbol{A}_0 \boldsymbol{X}^{\text{int}} + \boldsymbol{b}_0) + \boldsymbol{b}_1, \boldsymbol{H}_{11} = f_1(\boldsymbol{H}_1) = \boldsymbol{A}_1(\boldsymbol{A}_1 \boldsymbol{X}^{\text{int}} + \boldsymbol{b}_1) + \boldsymbol{b}_1.$$

Then

$$\begin{aligned}
\boldsymbol{E}_2 &= \boldsymbol{p}_{00} \boldsymbol{H}_{00} + \boldsymbol{p}_{01} \boldsymbol{H}_{01} + \boldsymbol{p}_{10} \boldsymbol{H}_{10} + \boldsymbol{p}_{11} \boldsymbol{H}_{11} \\
&= (\boldsymbol{p}_{00} \boldsymbol{A}_{00} + \boldsymbol{p}_{01} \boldsymbol{A}_{01} + \boldsymbol{p}_{10} \boldsymbol{A}_{10} + \boldsymbol{p}_{11} \boldsymbol{A}_{11}) \boldsymbol{X}^{\text{int}} \\
&+ (\boldsymbol{p}_{00} \boldsymbol{A}_0 \boldsymbol{b}_0 + \boldsymbol{p}_{01} \boldsymbol{A}_0 \boldsymbol{b}_1 + \boldsymbol{p}_{10} \boldsymbol{A}_1 \boldsymbol{b}_0 + \boldsymbol{p}_{11} \boldsymbol{A}_1 \boldsymbol{b}_1) + (\boldsymbol{p}_{00} \boldsymbol{b}_0 + \boldsymbol{p}_{01} \boldsymbol{b}_0 + \boldsymbol{p}_{10} \boldsymbol{b}_1 + \boldsymbol{p}_{11} \boldsymbol{b}_1),
\end{aligned}$$

where $\boldsymbol{A}_\mathbf{i} = \boldsymbol{A}_{i_1} \boldsymbol{A}_{i_2}, \forall \mathbf{i} = (i_1, i_2) \in \Omega_2$.

**The $n$-th iteration ($|\mathbf{i}| = n$).** Inductively, we have

$$\mathbb{H}^{(n)} = \{\boldsymbol{H}_\mathbf{i}\}_{|\mathbf{i}|=n}, \ \boldsymbol{H}_\mathbf{i} = f_\mathbf{i}(\boldsymbol{X}^{\text{int}}), \ \boldsymbol{E}_n = \sum_{|\mathbf{i}|=n} \boldsymbol{p}_\mathbf{i} \boldsymbol{H}_\mathbf{i}. \tag{3}$$

Note that $\boldsymbol{H}_\mathbf{i} \in \mathbb{R}^{N \times H}$ and each column of $\boldsymbol{H}_\mathbf{i}$ is a point in $\mathbb{R}^N$, so we regard it as a subset of $\mathbb{R}^N$ with $H$ elements. Thus $\mathbb{H}^{(n)}$ is a subset of $\mathbb{R}^N$ with $H \times 2^n$ elements (including duplicate elements). Because of Theorem 4.1, we call $\mathbb{H}^{(n)}$ the *fractal representation* with depth $n$ of notes.

### 3.3 Representation Layer

After $n$ iterations of IFS layer, the dynamic trajectory of IFS is obtained:

$$\mathcal{O} = \{\boldsymbol{E}_1, \boldsymbol{E}_2, ..., \boldsymbol{E}_n\}.$$

In general, the global representation $\boldsymbol{R}$ of notes is obtained by time average or concatenation operations on $\mathcal{O}$.

$$\boldsymbol{R} = \frac{1}{n}\sum_{i=1}^{n}\boldsymbol{E}_i \in \mathbb{R}^{N \times H} \quad \text{or} \quad \boldsymbol{R} =\|_{i=1}^{n} \boldsymbol{E}_i \in \mathbb{R}^{N \times nH},$$

where $\|$ is the concatenation operator. Because of Theorem 4.2, we call $\boldsymbol{R}$ the *ergodic representation* of notes. In practice, we adopt weighted time average or weighted concatenation. According to the Theorem C.1, we use heuristic weights (Here, we understand it as the average expansion factor of the distance between two points after affine transformation). Let $r = \sqrt{\ln(N) + \gamma}$, where $\gamma \approx 0.577215664$ is the Euler constant. Suppose $\boldsymbol{r} = (r_1, r_2, ..., r_n)$ is a learnable n-dimensional vector with initial value $r_i = \left(\frac{1}{r}\right)^{i-1}$. Then the ergodic representation of notes is

$$\boldsymbol{R} = \sum_{i=1}^{n} r_i\boldsymbol{E}_i \in \mathbb{R}^{N \times H} \quad \text{or} \quad \boldsymbol{R} =\|_{i=1}^{n} r_i\boldsymbol{E}_i \in \mathbb{R}^{N \times nH}.$$

### 3.4 Output Layer

Let $\boldsymbol{W}^{\text{out}} \in \mathbb{R}^{H \times P}$ be a learnable parameter matrix, where $P$ is the dimension of the output layer (such as the number of class labels). If $\boldsymbol{R}$ is generated by $\mathcal{O}$ concatenation, then let $\boldsymbol{W}^{\text{out}} \in \mathbb{R}^{nH \times P}$. There are two ways to construct output layer, one is to use a Single-Layer Perception (SLP) as output, that is

$$\boldsymbol{O} = \boldsymbol{R}\boldsymbol{W}^{\text{out}} + \boldsymbol{b}_{\text{out}};$$

the other is to use $f_0, f_1$ for *Mixed Propagation* (MP), that is, let $\boldsymbol{R}_0 = f_0(\boldsymbol{R}\boldsymbol{W}^{\text{out}})$ and $\boldsymbol{R}_1 = f_1(\boldsymbol{R}\boldsymbol{W}^{\text{out}})$, where the biases of $f_0, f_1$ are removed, then the output

$$\boldsymbol{O} = p_0\boldsymbol{R}_0 + p_1\boldsymbol{R}_1 + \boldsymbol{b}_{\text{out}}.$$

Where the bias $\boldsymbol{b}_{\text{out}} \in \mathbb{R}^P$ is an optional learnable parameter vector.

### 3.5 Initialization of Learnable Variables

The learnable parameters of IGNNS include input layer matrix $\boldsymbol{W}^{\text{int}} \in \mathbb{R}^{F \times H}$, adjoint probability vector $\boldsymbol{p} = (p_0, p_1) \in \mathbb{R}^2$ of IFS, biases $\boldsymbol{b}_0, \boldsymbol{b}_1 \in \mathbb{R}^H$ of IFS layer, weight coefficient $\boldsymbol{r} = (r_1, r_2, ..., r_n)$ of representation layer, matrix $\boldsymbol{W}^{\text{out}} \in \mathbb{R}^{H \times P}$ of output layer and bias $\boldsymbol{b}_{\text{out}} \in \mathbb{R}^P$ of output layer. Among them, $\boldsymbol{W}^{\text{int}}$ and $\boldsymbol{W}^{\text{out}}$ are the required learnable parameters, using the initialization described in Glorot & Bengio (2010); $\boldsymbol{b}_0$, $\boldsymbol{b}_1$ and $\boldsymbol{b}_{\text{out}}$ are optional learnable parameters with an initial value $\boldsymbol{0}$; $\boldsymbol{p}$ is a optional learnable parameter, for undirected graph, setting $p_0 \in [0.5 - 0.1, 0.5 + 0.1]$, and for directed graph, setting (For the reasons, see Appendix G)

$$p_0 = \frac{\det \boldsymbol{D}_1}{\det \boldsymbol{D}_0 + \det \boldsymbol{D}_1} \quad , \quad p_1 = \frac{\det \boldsymbol{D}_0}{\det \boldsymbol{D}_0 + \det \boldsymbol{D}_1};$$

$\boldsymbol{r}$ is a optional learnable parameter, and its initial value as defined in 3.3. Let $n$ be the number of iterations of IFS. We regard $n$ as the depth of IGNNS. Thus, IGNNS is denoted as

$$\boldsymbol{O} = \text{IGNNS}(\boldsymbol{X}, \boldsymbol{A}; \boldsymbol{W}^{\text{int}}, n, \boldsymbol{p}, \boldsymbol{b}_0, \boldsymbol{b}_1, \boldsymbol{r}, \boldsymbol{W}^{\text{out}}, \boldsymbol{b}_{\text{out}}) \quad \text{or simply} \quad \boldsymbol{O} = \text{IGNNS}(\boldsymbol{X}, \text{IFS}),$$

where IFS is induced by relational matrix $\boldsymbol{A}$. The output of IGNNS can be used as the input of downstream tasks, and can also be connected to other network architectures.

### 3.6 Theoretical Time Complexity of IGNNS

Let $n, N, H, P$ be defined as above. Let $T(\cdot)$ denote the number of calculations of an object. For input layer,

$$T(\text{input layer}) = NFH + NH = O(NFH).$$

For IFS layer, during the iterative calculation, we will store the previous calculation results, thus $T(\mathbb{H}^{(1)}) = 2N^2H, T(\mathbb{H}^{(i)}) = 2 \times T(\mathbb{H}^{(i-1)}), i = 2, 3, ..., n$. It follows that $T(\mathbb{H}^{(i)}) = 2^i N^2 H, i = 1, 2, ..., n$. Similarly, $T(\{\boldsymbol{p_i}|||\mathbf{i}| = i\}) = 2i, i = 1, 2, ..., n$. Complete the above calculation, it is easy to see that $T(\boldsymbol{E}_i) = 2^i N H$. Thus

$$T(\text{IFS layer}) = \sum_{i=1}^{n} \left( T(\mathbb{H}^{(i)}) + T(\{\boldsymbol{p_i}|||\mathbf{i}| = i\}) + T(\boldsymbol{E}_i) \right) = O(2^n N^2 H).$$

It is easy to verify that $T(\text{representation layer}) = O(nNH)$. We assume that the output layer is constructed by mixed propagation, then $T(\text{output layer}) = O(N^2P)$ if $\boldsymbol{W}^{\text{out}} \in \mathbb{R}^{H \times P}$, and $T(\text{output layer}) = O(N^2P + nNHP)$ if $\boldsymbol{W}^{\text{out}} \in \mathbb{R}^{nH \times P}$. Then

$$T(\text{IGNNS}) = O(2^n N^2 H + N^2 P) \quad \text{or} \quad O(2^n N^2 H + N^2 P + nNHP).$$

In practice, for large graphs, $2^n N^2 H \gg N^2 P \gg nNHP$, thus $2^n N^2 H$ is the main factor affecting the time complexity of IGNNS. Furthermore, for large graphs of the same size, $n$ is the main important factor affecting time complexity. For citation network datasets such as citeser, Cora and PubMed, we suggest that $n \leq 8$ (see Appendix B).

## 4 GEOMETRIC PROPERTIES OF IGNNS

The discussion here assumes that affine $f_0$, $f_1$ are contractive. Otherwise, let $f_0 : x \to \frac{1}{\|\boldsymbol{A}_0\|_F + 1} \boldsymbol{A}_0 x + b_0$ and $f_1 : x \to \frac{1}{\|\boldsymbol{A}_1\|_F + 1} \boldsymbol{A}_1 x + b_1$. In practice, IGNNS does not use contractive affine. If contractive affine is used in IGNNS, it can be seen from the following theorems that the characterization ability of IGNNS decreases with the increase of IFS iterations, which is similar to the performance of Graph Convolution Network (GCN). Such a phenomenon is called over-smoothing (Li et al., 2018b; Xu et al., 2019; Chen et al., 2020), which suggests that as the number of layers increases, the representations of the nodes in GCN are inclined to converge to a certain value and thus become indistinguishable. In order to overcome the over-smoothing problem of deep GNN, people need to use some new methods such as Skip connection (Xu et al., 2018), Drop edge (Rong et al., 2020), Residual connection (Klicpera et al., 2019a), Identity mapping (Chen et al., 2020), Generalized message aggregation functions (Li et al., 2020) and so on. Generally speaking, deep network may lead to the decrease of generalization performance. To analyze which type of deep GNN would achieve better generalization performance, Xu et al. (2020) proposes a guiding theoretical framework.

**Theorem 4.1 (Fractal generation)** *Let $\mathbb{H}^{(n)} = \{\boldsymbol{H_i}\}_{|i|=n}$, which is a subset of $\mathbb{R}^N$ with $H \times 2^n$ elements (including duplicate elements), then*

$$d_H(\mathbb{H}^{(n)}, \mathbb{F}) \to 0, \quad n \to \infty,$$

*where $d_H$ is the Hausdorff distance defined on $\mathcal{H}(\mathbb{R}^N)$, the set of all nonempty compact subsets of $\mathbb{R}^N$, and $\mathbb{F}$ is the fractal set of IFS in IGNNS. In other words, as the number of iterations increases, $\mathbb{H}^{(n)}$ will be independent of node feature $\boldsymbol{X}$, only related to the graph structure described by $\boldsymbol{A}$.*

Let $T$ be the Hutchinson operator on $\mathcal{H}(\mathbb{R}^N)$, defined as $T(B) = f_0(B) \bigcup f_1(B), \forall B \in \mathcal{H}(\mathbb{R}^N)$. Then the **updated rule** of $\mathbb{H}^{(n)}$ satisfies

$$\mathbb{H}^{(n)} = T(\mathbb{H}^{(n-1)}) = \cdots = T^n(\mathbb{H}^{(0)}),$$

where $\mathbb{H}^{(0)} = \{\boldsymbol{X}^{\text{int}}\}$ is a subset of $\mathbb{R}^N$ with $H$ elements. In fractal geometry, $\mathbb{H}^{(n)}$ is used to draw the fractal image on the plane. First, taking initial value $\mathbb{H}^{(0)} = \{x_0\}$, where $x_0$ is a point in plane. For enough $n$, printing all the points of $\mathbb{H}^{(n)}$ on the screen to obtain the approximate fractal image.

**Theorem 4.2 (Ergodic property)** *Let $\boldsymbol{E}_n = \sum_{|i|=n} \boldsymbol{p_i H_i}$ be the mathematical expectation of $\mathbb{H}^{(n)} = \{\boldsymbol{H_i}\}_{|i|=n}$, then $\boldsymbol{E}_n$ converges to a constant matrix $\boldsymbol{E} \in \mathbb{R}^{N \times H}$ in Frobenius norm, i.e. $\lim_{n\to\infty} \boldsymbol{E}_n = \boldsymbol{E}$, where $\boldsymbol{E}[i, :] = (e_i, e_i, ..., e_i)^\top \in \mathbb{R}^H$ and $e_i \in \mathbb{R}$ is a constant, $i = 1, 2, ..., N$. Furthermore, the time average of the dynamic trajectory $\mathcal{O}$ of IFS satisfies*

$$\lim_{n\to\infty} \frac{1}{n} \sum_{i=1}^{n} \boldsymbol{E}_i = \lim_{n\to\infty} \boldsymbol{E}_n = \boldsymbol{E}$$

*and series $\sum_{i=1}^{\infty} r_i \boldsymbol{E}_i \in \mathbb{R}^{N \times H}$ converges in Frobenius norm.*

Table 1: Summary statistics of the benchmark datasets used in the experiment.

| Dataset | Nodes | Edges | Features | Classes | Training | Validation | Testing |
|---------|-------|-------|----------|---------|----------|------------|---------|
| Cora | 2708 | 5429 | 1433 | 7 | 140 | 500 | 1000 |
| Citeseer | 3327 | 4732 | 3703 | 6 | 120 | 500 | 1000 |
| Pubmed | 19717 | 44338 | 500 | 3 | 60 | 500 | 1000 |

Theorem 4.2 shows that as long as the number of iterations is large enough, the embeddings of nodes will be close to linear correlation, and the representation ability of IGNNS will decline. However, in the framework of IGNNS, because the spectral radius $\rho(\boldsymbol{A}_0) = \rho(\boldsymbol{A}_1) = 1$, IFS is not contractive in general, and IGNNS still has the ability of depth feature representation.

## 5 EXPERIMENTS

### 5.1 EXPERIMENTAL TASK: SEMI-SUPERVISED NODE CLASSIFICATION

Let $\boldsymbol{Z} = \text{softmax}(\boldsymbol{O})$, where $\boldsymbol{Z} \in \mathbb{R}^N$ and $\text{softmax}(\cdot)$ is the softmax activation function, defined as $\text{softmax}(x_i) = \frac{1}{\mathcal{Z}} \exp(x_i)$ with $\mathcal{Z} = \sum_i \exp(x_i)$, is applied row-wise. For semi-supervised multiclass classification, we employ the following cross-entropy to evaluate error over all labeled examples: $L = -\sum_{l \in \mathbb{Y}_L} \sum_{i=1}^{P} \boldsymbol{Y}[l,i] \ln \boldsymbol{Z}[l,i]$, where $\mathbb{Y}_L$ is the set of node indices that have labels with $P$ classes, $\boldsymbol{Y}[l,:]$ is a one-hot vector of size $P$ representing the class of node $l$ and $\boldsymbol{Z}[l,:]$ is the row $l$ of the matrix $\boldsymbol{Z}$.

### 5.2 EXPERIMENTAL SETUP

**Datasets.** In our experiment, we use three standard citation network benchmark datasets for evaluation, including Cora, Citeseer, Pubmed and apply the standard fixed training/validation/testing split (Yang et al., 2016; Kipf & Welling, 2017; Veličković et al., 2018) on above datasets, with 20 nodes per class for training, 500 nodes for validation and 1,000 nodes for testing. In these citation networks, papers are represented as nodes, and citations of one paper by another are denoted as edges. Node features are the bag-of-words vector of papers, and node label is the only one academic topic of a paper. See Table 1 for more details.

**Parameter Setting.** Random seed for Tensorflow and Numpy is set to 1234. ReLU (Nair & Hinton, 2010) is used as the activation function in input layer and output layer. Dropout (Srivastava et al., 2014) is applied to input layer, IFS layer and output layer. In representation layer, we adopt weighted time average to get the global representation of notes. In output layer, we adopt the method of mixed propagation to get the output of IGNNS. We use the AdamOptimizer (P.Kingma & Ba, 2015) during training. More details of hyper-parameters are shown in Table 2. During training stage, we select the best model to maximize the accuracy of the validation set and use early stopping with a patience of 100 epochs.

### 5.3 EXPERIMENTAL RESULT

We compare with those models that strictly follow the standard of experiment setup of semi-supervised node classification, i.e. the standard fixed training/validation/testing split (Yang et al., 2016; Kipf & Welling, 2017) is applied on dataset. For baselines, we include recent deep GNN models such as JKNet (Xu et al., 2018), APPNP (Klicpera et al., 2019a), Attention-based models such as GAT (Veličković et al., 2018), AGNN (Thekumparampil et al., 2018) and H-GAT (Gulcehre et al., 2019) , and other models such as TAGCN (Du et al., 2017) and N-GCN (Abu-El-Haija et al., 2018). We also include three state-of-the-art shallow GNN models: Planetoid (Yang et al., 2016), GCN (Kipf & Welling, 2017) and DGCN (Zhuang & Ma, 2018). The detailed results are shown in Table 3.

We can see from Table 3 that the improved performance of model IGNNS in dataset Cora and Citeseer is much higher than that in dataset Pubmed. To understand why this happens, we analyze

Table 2: hyper-parameters in experiment.

| Setting | Cora | Citeseer | Pubmed |
|---|---|---|---|
| Neighborhood Normalization | symmetric | mean-pooling | symmetric |
| Learning rate | 0.005 | 0.002 | 0.01 |
| Initial value of $p_0$ | 0.5 | 0.5 | 0.5 |
| Dropout | 0.9 | 0.9 | 0.8 |
| Weight decay | 5e-3 | 5e-3 | 5e-3 |
| Epochs | 1000 | 1000 | 1000 |
| Hidden dimensions | 48 | 72 | 72 |
| Number of iterations in IFS layer | 5 | 4 | 4 |
| Learnable adjoint probability vector | False | True | False |
| Learnable representation layer coefficient | True | True | True |
| Use bias for IFS layer | True | False | False |
| Use bias for output layer | False | False | False |

Table 3: Summary of classification accuracy (%) results on Cora, Citeseer and Pubmed. The results are taken from the corresponding papers. The first value in brackets indicates the total training time in seconds and the second value in brackets indicates the average training time in seconds per epoch.

| Method | Cora | Citeseer | Pubmed |
|---|---|---|---|
| Planetoid (Yang et al., 2016) | 75.7 | 64.7 | 77.2 |
| GCN (Kipf & Welling, 2017) | 81.5 | 70.3 | 79.0 |
| GAT (Veličković et al., 2018) | 83.0 | 72.5 | 79.0 |
| TAGCN (Du et al., 2017) | 83.3 | 71.4 | **81.1** |
| JKNet (Xu et al., 2018) | 81.1 | 69.8 | 78.1 |
| AGNN (Thekumparampil et al., 2018) | 83.1 | 71.7 | 79.9 |
| N-GCN (Abu-El-Haija et al., 2018) | 83.0 | 72.2 | 79.5 |
| DGCN (Zhuang & Ma, 2018) | 83.5 | 72.6 | 80.0 |
| APPNP (Klicpera et al., 2019a) | 83.3 | 71.8 | **81.1** |
| H-GAT (Gulcehre et al., 2019) | 83.5 | 72.9 | − |
| **IGNNS** (ours) | **86.3**(44s, 0.17s) | **75.1**(65s, 0.16s) | 80.5(221s, 1.47s) |

the characteristics of these citation networks. We consider two statistical properties of networks, one is the Network Density $d(\mathcal{G})$, which is defined as $d(\mathcal{G}) = \frac{2L}{N(N-1)}$, where $N$ is the number of nodes and $L$ is the number of edges, and the other is the Average Clustering Coefficient $C$, which is defined as $C = \frac{1}{N} \sum_{i \in V} C_i$, where $V$ is the set of nodes, $C_i = \frac{2e_i}{k_i(k_i-1)}$, $k_i$ is the number of the neighbors of node $v_i$ and $e_i$ is the number of undirected edges between $k_i$ neighbors. The small Network Density means the strong global sparsity of the network, and the small Average Clustering Coefficient means the strong sparsity of the neighbors of nodes. The calculation results of the statistical characteristics of the network are shown in Table 4. We can see from Table 4 that Pubmed is more sparse than Cora and Citeseer. The performance of IGNNS benefits from the bidirectional mixed propagation of information between nodes, but this sparsity weakens the gain of IGNNS.

## 5.4 PERFORMANCE OF COMPLETELY LINEAR IGNNS

In Nonlinear IGNNS, we use the nonlinear activation function ReLU$(x)$, learn adjoint probability vector $\boldsymbol{p} = (p_0, p_1)$ by $p_0 \leftarrow \frac{\text{ReLU}(p_0)+0.1}{\text{ReLU}(p_0)+\text{ReLU}(p_1)+0.2}, p_1 \leftarrow \frac{\text{ReLU}(p_1)+0.1}{\text{ReLU}(p_0)+\text{ReLU}(p_1)+0.2}$ and learn the representation layer coefficient $\boldsymbol{r} = (r_1, r_2, ..., r_n)$ by $r_i \leftarrow \text{ReLU}(r_i)$ with initial value $r_i = \left(\frac{1}{r}\right)^{i-1}$ where $r = \sqrt{\ln(N) + 0.577215664}$. In this experiment, to get a completely linear IGNNS, we let all the activation functions be the identity function, i.e. $\sigma(x) = x$, and let the adjoint probability vector $\boldsymbol{p} = (p_0, p_1)$ and the representation layer coefficient $\boldsymbol{r} = (r_1, r_2, ..., r_n)$ be hyperparameters

Table 4: Statistical characteristics of the networks. Bold for minimum.

| Statistical characteristics | Cora | Citeseer | Pubmed |
|---|---|---|---|
| Network Density | 0.00144000 | 0.00084514 | **0.00022805** |
| Average Clustering Coefficient | 0.24067330 | 0.14147102 | **0.06017521** |

Table 5: Performance of completely linear IGNNS on Cora, Citeseer and Pubmed.

| Method | Cora | Citeseer | Pubmed |
|---|---|---|---|
| GCN (Kipf & Welling, 2017) | 81.5 | 70.3 | 79.0 |
| IGNNS(Linear) | **83.9** | **72.4** | **79.9** |

without learning. For Citeseer, we let $p_0 = 0.6$, and for Pubmed, we use bias for IFS layer. Except for the above changes, experimental task and the other experimental setups remain unchanged as showed in section 5.1 and 5.2 respectively. We can see from Table 5 that the performance of completely linear IGNNS is better than that of baseline model GCN. Compared with other models, completely linear IGNNS is still competitive. This is due to the fact that the IFS can extract more features than spectral filters. For more discussion, Let $A$ be the normalization adjacency matrix of graph $\mathcal{G}$, and let $A_0$ and $A_1$ be defined as in section 3.2. We further assume that the dimension of the hidden space is equal to 1. This means that the input of GNN (GCN or IGNNS) is a point $x_0 = X^{\text{int}} = XW^{\text{int}} \in \mathbb{R}^{N \times 1}$. Let $n$ be the depth of GNN, for IGNNS, equal to the number of iterations of IFS. For convenience, we ignore the activation function and parameter matrix. Let $f(x) = Ax + b$, $f_0(x) = A_0 x + b_0$ and $f_1(x) = A_1 x + b_1$. For GCN, the message passing results are

$$\{f(x_0)\}, \{f^2(x_0)\}, ..., \{f^n(x_0)\}.$$

We see that each iteration only gets one value, i.e. $|\{f^n(x_0)\}| = 1$. For IGNNS, the message passing results are as follows:

$$\{f_0(x_0), f_1(x_0)\}, \{f_0(f_0(x_0)), f_0(f_1(x_0)), f_1(f_0(x_0)), f_1(f_1(x_0))\}, ..., \{f_{\mathbf{i}}(x_0)\}_{|\mathbf{i}|=n}.$$

If $f_0$ and $f_1$ satisfy separation condition, i.e. $f_0(x) = f_1(y)$ implies $x = y$, then $|\{f_{\mathbf{i}}(x_0)\}_{|\mathbf{i}|=n}| = 2^n$. This means that IGNNS can extract more information than GCN. Even if $f_0$ and $f_1$ are contractive mappings, by Theorem 4.1, we have $\{f_{\mathbf{i}}(x_0)\}_{|\mathbf{i}|=n} \to \mathbb{F}$, where $\mathbb{F}$ is the fractal set of IFS induced by $f_0$ and $f_1$. Generally speaking, $\mathbb{F}$ is a uncountable compact set. This means that when $n$ is large, the features may still be distinguishable.

## 6 CONCLUSION

In this paper, we propose a new framework of graph neural networks, IGNNS, which give a connection between Graph Neural Networks and Iterated Function System. We use IFS to simulate the bidirectional message passing process of graph neural network, and obtain the fractal representation and ergodic representation of graph nodes, which are very helpful for downstream tasks. The experiments show that we have achieved good results in semi-supervised node classification task. Interesting directions for future work include pruning the iterative path space $\{0, 1\}^n$ to reduce the computational complexity, coding graph structured data with IFS, and establishing more interesting connections between IFS and graph neural networks.

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

## A  THE FRACTAL REPRESENTATION OF GRAPH $\mathcal{G}$ WITH ONLY ONE SELF ADJACENT NODE $v$.

For the sake of discussion, we assume that the dimension of the hidden space is equal to 1. We assume that messages are sent from node $v$, propagate in two directions (clockwise and anticlockwise), and are finally received by node $v$. The received messages in the clockwise direction become one-third of the original, and the received messages in the anticlockwise direction become one-third of the original plus a constant of $\frac{2}{3}$. It is expressed by mathematical formula as follows

$$f_0(x) = \frac{x}{3}, \quad f_1(x) = \frac{x}{3} + \frac{2}{3}, \quad x \in \mathbb{R}.$$

For Bi-GCN, messages are delivered independently in both directions. In other words, there are two independent channels, and the message passing (transmitting or receiving) can only be carried out by their own channels. Let $x_0 \in \mathbb{R}$ be the initial message. In the clockwise direction, after $n$ passes, the received messages are

$$\frac{x_0}{3}, \frac{x_0}{3^2}, ..., \frac{x_0}{3^n} \to 0.$$

In the anticlockwise direction, the received messages are

$$\frac{x_0}{3} + \frac{2}{3}, \frac{x_0}{3^2} + \frac{2}{3^2} + \frac{2}{3}, ..., \frac{x_0}{3^n} + \sum_{i=1}^{n} \frac{2}{3^i} \to 1.$$

For IGNNS, the two channels have a connection point at node $v$. First, node $v$ sends the message $x_0$ in both directions, and the connection point of node $v$ will receive two messages $\{f_0(x_0), f_1(x_0)\}$. In the second launch, any message ($f_0(x_0)$ or $f_1(x_0)$) can be sent in both directions, so the received messages are

$$\{f_0(f_0(x_0)), f_0(f_1(x_0)), f_1(f_0(x_0)), f_1(f_1(x_0))\}.$$

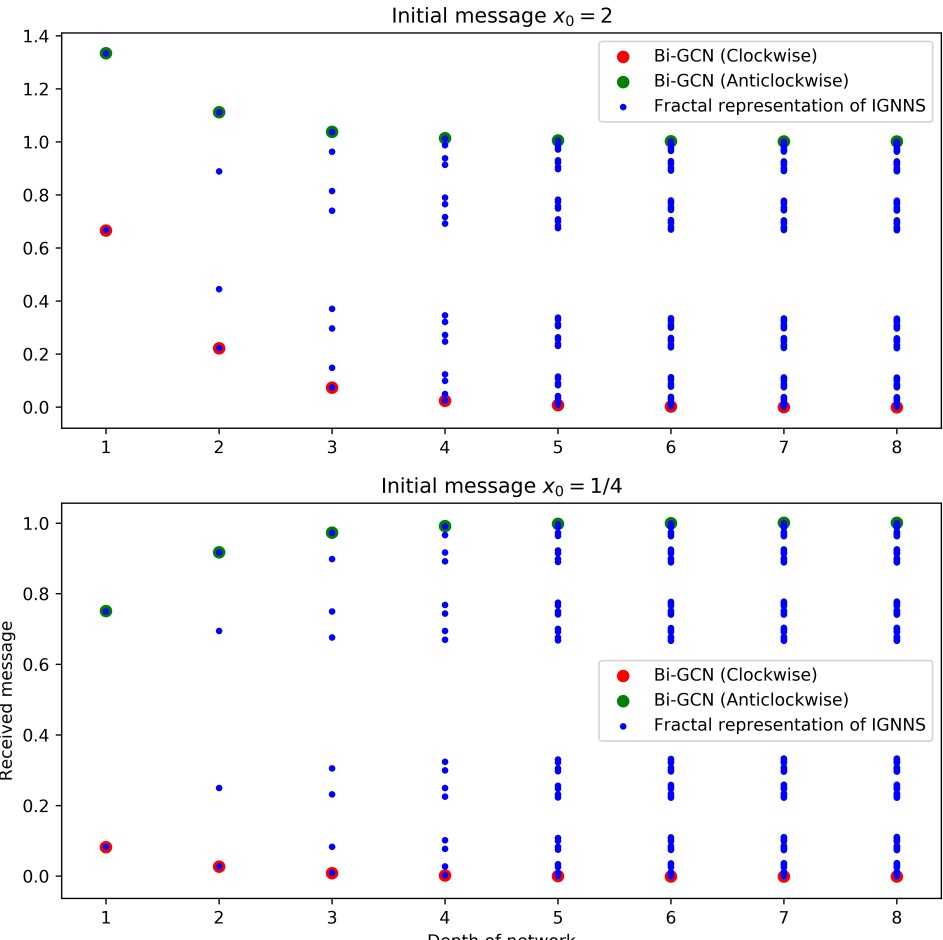

Figure 3: Comparison of feature extraction ability between Bi-GCN and IGNNS. Bi-GCN gets boundary messages and IGNNS gets all messages.

In summary, After $n$ passes, the received messages are

$$\mathbb{H}^{(n)} = \{f_{\mathbf{i}}(x_0)\}_{|\mathbf{i}|=n} \to C,$$

where $C$ is the famous Cantor Set. This means that we have not only received one message, but $2^n$, since $f_0$ and $f_1$ satisfy the separation condition. We can see from Figure 3 that Bi-GCN gets boundary messages and IGNNS gets all messages. Let $\boldsymbol{p} = (p_0, p_1)$ be the adjoint probability vector, then the mathematical expectation $\boldsymbol{E}_n$ of $\mathbb{H}^{(n)}$ is $\sum_{|\mathbf{i}|=n} \boldsymbol{p_i} f_{\mathbf{i}}(x_0)$. We interpret $\boldsymbol{E}_n$ as the average of all messages received. Now the question is, fractal representation gets enough messages, but is there redundancy in these messages? How to select the valid message from the fractal representation becomes the focus of our research in the next stage.

## B  ANALYSIS OF TIME COMPLEXITY ON CORA, CITESEER AND PUBMED.

From section 3.6, we have known that the time complexity in Experiment 5.1 is $O(2^n N^2 H + N^2 P)$, where $n$ is the number of iterations of IFS (the depth of IGNNS), $N$ is the number of nodes, $H$ is the dimension of the latent space and $P$ is the dimension of the output layer. In this section, we compare the real running time (100 epochs) of IGNNS on Cora, Citeseer and Pubmed. Let $H = 8$, then $2^n N^2 H$ is the main factor affecting the time complexity of IGNNS. Let the depth of IGNNS

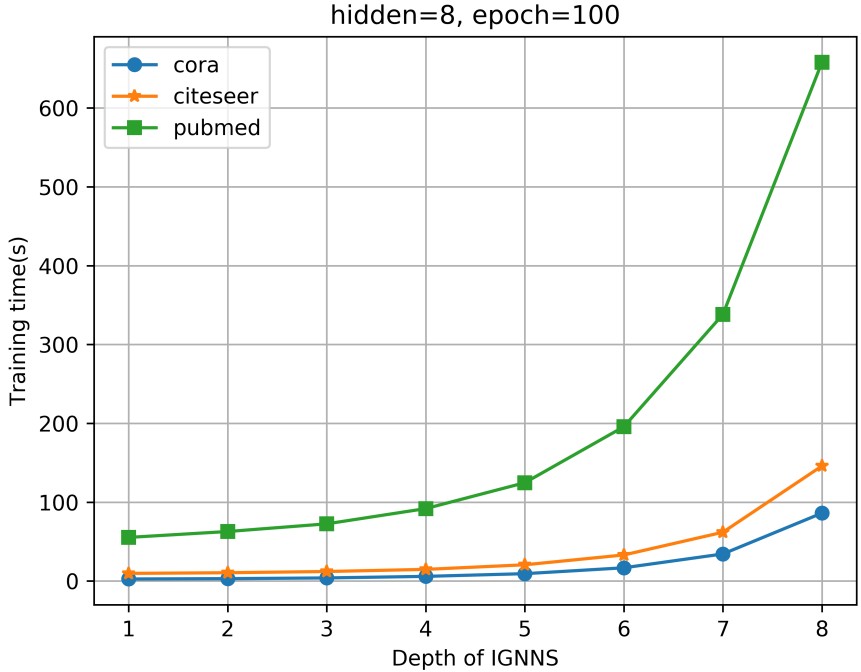

Figure 4: Real training time on Cora, Citeseer and Pubmed.

go from 1 to 8, and compare the real training time on Cora, Citeseer and Pubmed. We can see from Figure 4 that the actual results are basically consistent with the theoretical analysis results.

## C    FROBENIUS NORM OF MATRIX

**Theorem C.1** *Let $\boldsymbol{A} \in \mathbb{R}^{N \times N}$ be the adjacency matrix of a graph with no weights, i.e. $\boldsymbol{A}_{i,j} = 1$ if there exists an edge $i \to j$ in the graph and $\boldsymbol{A}_{i,j} = 0$ otherwise, and $\boldsymbol{A}_1 = \boldsymbol{D}_1^{-1} tril(\boldsymbol{A})$ (or $\boldsymbol{A}_1 = \boldsymbol{D}_1^{-\frac{1}{2}} tril(\boldsymbol{A}) \boldsymbol{D}_1^{-\frac{1}{2}}$) as defined in IGNNS, then*

$$\|\boldsymbol{A}_1\|_F \geq \sqrt{\ln(N) + \gamma},$$

*where $\gamma \approx 0.577215664$ is the Euler constant.*

**Proof.   Case 1:** $\boldsymbol{A}_1 = \boldsymbol{D}_1^{-1} tril(\boldsymbol{A})$. Let $tril(\boldsymbol{A}) = (a_{ij})_{N \times N}$, $\boldsymbol{D}_1 = diag(d_1, d_2, ..., d_N)$ be the degree matrix of $tril(\boldsymbol{A})$ and $\boldsymbol{A}_1 = (b_{ij})_{N \times N}$. Note that $tril(\boldsymbol{A})$ is an lower triangular matrix, then $d_i = \sum_{j=1}^{N} a_{ij} = \sum_{j=1}^{i} a_{ij}$. Since $a_{ij} \in \{0, 1\}$, we have $i \geq d_i$. Computing the Frobenius Norm of $\boldsymbol{A}_1$ as follows:

$$\|\boldsymbol{A}_1\|_F^2 = \sum_{i=1}^{N} \sum_{j=1}^{N} b_{ij}^2 = \sum_{i=1}^{N} \sum_{j=1}^{i} b_{ij}^2 = \sum_{i=1}^{N} \sum_{j=1}^{i} \left( \frac{a_{ij}}{d_i} \right)^2. \tag{4}$$

$\forall i \in \{1, 2, ..., N\}$, note that $d_i$ elements in $\{a_{ij}\}_{j=1}^{i}$ are 1 and the rest are 0. It follows that

$$\sum_{j=1}^{i} \left( \frac{a_{ij}}{d_i} \right)^2 = \left( \frac{1}{d_i} \right)^2 \times \sum_{j=1}^{i} a_{ij}^2 = \left( \frac{1}{d_i} \right)^2 \times d_i = \frac{1}{d_i} \geq \frac{1}{i}. \tag{5}$$

It follows from (4) and (5) that

$$\|\boldsymbol{A}_1\|_F^2 \geq \sum_{i=1}^{N} \frac{1}{i}.$$

So $\|\boldsymbol{A}_1\|_F \geq \sqrt{\sum_{i=1}^{N} \frac{1}{i}} \approx \sqrt{\ln(N) + \gamma}$, where $\gamma \approx 0.577215664$ is the Euler constant.

**Case 2:** $\boldsymbol{A}_1 = \boldsymbol{D}_1^{-\frac{1}{2}} tril(\boldsymbol{A}) \boldsymbol{D}_1^{-\frac{1}{2}}$. Computing the Frobenius Norm of $\boldsymbol{A}_1$ as follows:

$$\|\boldsymbol{A}_1\|_F^2 = \sum_{i=1}^{N} \sum_{j=1}^{N} b_{ij}^2 = \sum_{i=1}^{N} \sum_{j=1}^{i} b_{ij}^2 = \sum_{i=1}^{N} \sum_{j=1}^{i} \frac{a_{ij}^2}{d_i d_j}. \tag{6}$$

$\forall i \in \{1, 2, ..., N\}$, it follows from $j \geq d_j$ that

$$\sum_{j=1}^{i} \frac{a_{ij}^2}{d_i d_j} \geq \frac{1}{d_i} \sum_{j=1}^{i} \frac{1}{j} \cdot a_{ij}^2. \tag{7}$$

Note that $d_i$ elements in $\{a_{ij}\}_{j=1}^{i}$ are 1 and the rest are 0. It follows from Rearrangement inequality that

$$\sum_{j=1}^{i} \frac{1}{j} \cdot a_{ij}^2 \geq \frac{1}{i - d_i + 1} \cdot a_{i(i-d_i+1)}^2 + \frac{1}{i - d_i + 2} \cdot a_{i(i-d_i+2)}^2 + \cdots + \frac{1}{i} \cdot a_{ii}^2, \tag{8}$$

where $a_{i(i-d_i+1)} = a_{i(i-d_i+2)} = \cdots = a_{ii} = 1$. Thus

$$\sum_{j=1}^{i} \frac{1}{j} \cdot a_{ij}^2 \geq \frac{1}{i} \cdot 1^2 + \frac{1}{i} \cdot 1^2 + \cdots + \frac{1}{i} \cdot 1^2 = d_i \times \frac{1}{i}. \tag{9}$$

It follows from (6), (7) and (9) that

$$\|\boldsymbol{A}_1\|_F^2 \geq \sum_{i=1}^{N} \frac{1}{i}.$$

Which completes the proof.

# D  INTRODUCTION TO ITERATED FUNCTION SYSTEM

In order to prove Theorem 4.1 and Theorem 4.2, we will briefly introduce the relevant conclusions on IFS in this section, and we will not give the proof here. More details of IFS Theory can be found in Hutchinson (1981); Elton (1987); Barnsley (1988); Falconer (1990); Massopust (2017).

## D.1  FRACTAL SPACE

Let $(X; d)$ be a complete metric space. Let $\mathcal{H}(X)$ denote a set consisting of all nonempty compact subsets of $X$. Hausdorff distanceon $d_H$ on $\mathcal{H}(X)$ defined by

$$d_H(A, B) = \max\{\max_{a \in A} \min_{b \in B} d(a, b), \max_{b \in B} \min_{a \in A} d(a, b)\}, \forall A, B \in \mathcal{H}(X). \tag{10}$$

**Theorem D.1** $(\mathcal{H}(X); d_H)$ *is a complete metric space.*

We call $(\mathcal{H}(X); d_H)$ a Fractal space. Let $\{f_i\}_{i=1}^{n}$ be a set of mappings on $(X; d)$. Hutchinson operator $T : (\mathcal{H}(X); d_H) \rightarrow (\mathcal{H}(X); d_H)$ defined as

$$T(B) = \bigcup_{i=1}^{n} f_i(B), \forall B \in \mathcal{H}(X). \tag{11}$$

**Theorem D.2** *If* $\{f_i\}_{i=1}^{n}$ *is a set of contractive mappings on* $(X; d)$*, then Hutchinson operator* $T$ *is a contractive mapping on* $(\mathcal{H}(X); d_H)$*.*

### D.2    MARKOV OPERATOR OF IFS

Let $(X; d)$ be a complete metric space. Let $\mathcal{M}(X)$ be the set of all probability measures on $X$. Let $C(X)$ be the set of all continuous functions mapping $X$ to $\mathbb{R}$. We say that $f \in \widetilde{Lip1}$, if $|f(x) - f(y)| \leq d(x, y), \forall x, y \in X$. It is easy to see that if $f \in \widetilde{Lip1}$ then $f \in C(X)$. Hutchinson metric $d_M$ on $\mathcal{M}$ defined as

$$d_M(\mu, \nu) = \sup \left\{ \left| \int_X f d\mu - \int_X f d\nu \right| \, | f \in \widetilde{Lip1} \right\}, \forall \mu, \nu \in \mathcal{M}(X). \tag{12}$$

**Theorem D.3**  $(\mathcal{M}(X); d_M)$ *is a complete metric space.*

Let IFS $= \{X; f_1, f_2, ..., f_n; \boldsymbol{p}\}$, the Markov operator $M : \mathcal{M}(X) \to \mathcal{M}(X)$ of IFS defined as

$$M\mu = \sum_{i=1}^{n} p_i \mu \circ f_i^{-1}, \mu \in \mathcal{M}(X). \tag{13}$$

**Theorem D.4** *Markov operator $M$ of IFS is a contractive mapping on space $(\mathcal{M}(X); d_M)$.*

Let measure sequence $\{\mu_i\} \subset \mathcal{M}$ and $\mu \in \mathcal{M}$, we call $\{\mu_i\}$ weakly convergent to $\mu$ if the following equation holds:

$$\lim_{i \to \infty} \int_X f d\mu_i = \int_X f d\mu, \forall f \in C(X), \tag{14}$$

denoted as $\mu_i \xrightarrow{w} \mu$ as $i \to \infty$.

**Theorem D.5** *If $\mu_i \xrightarrow{d_M} \mu$ as $i \to \infty$, then $\mu_i \xrightarrow{w} \mu$ as $i \to \infty$.*

### E    THE PROOF OF THEOREM 4.1

**Proof.**    By Theorem D.1, $(\mathcal{H}(\mathbb{R}^N); d_H)$ is a complete metric space. Let $T$ be the Hutchinson operator on $\mathcal{H}(\mathbb{R}^N)$, defined as $T(B) = f_0(B) \bigcup f_1(B), \forall B \in \mathcal{H}(\mathbb{R}^N)$. By Theorem D.2, $T$ is a contractive mapping on $(\mathcal{H}(\mathbb{R}^N); d_H)$. It follows from the Banach fixed point theorem that there exits a unique compact subset $\mathbb{F}$ of $\mathcal{H}(\mathbb{R}^N)$ such that

$$\mathbb{F} = T(\mathbb{F}) = f_0(\mathbb{F}) \bigcup f_1(\mathbb{F}),$$

which implies that $\mathbb{F}$ is the fractal set of IFS. Further more, $\forall B \in \mathcal{H}(\mathbb{R}^N)$, we have $T^n(B) \xrightarrow{d_H} \mathbb{F}$. The above convergence is independent of the choice of initial value. Thus, let $\mathbb{H}^{(0)} = \{\boldsymbol{X}^{\text{int}}\} = \{x_1, x_2, ..., x_H\}, x_i \in \mathbb{R}^N$, we have

$$\mathbb{H}^{(n)} = T(\mathbb{H}^{(n-1)}) = \cdots = T^n(\mathbb{H}^{(0)}) \xrightarrow{d_H} \mathbb{F}.$$

The above result indicates that when $n$ is large enough, $\mathbb{H}^{(n)}$ is close to the fractal set $\mathbb{F}$ of IFS in the sense of Hausdorff distance, and has nothing to do with the choice of initial value $\mathbb{H}^{(0)}$.

### F    THE PROOF OF THEOREM 4.2

**Proof.** It suffices to prove that $\forall j \in \{1, 2, ..., H\}$, $\boldsymbol{E}_n[:, j]$, the $j$ column of $\boldsymbol{E}_n$, satisfies

$$\lim_{n \to \infty} \boldsymbol{E}_n[:, j] = \begin{pmatrix} e_1 \\ e_2 \\ \vdots \\ e_N \end{pmatrix}_{N \times 1}.$$

For this purpose, let $x_j$ be the $j$ column of $\boldsymbol{X}^{\text{int}}$ as defined in (2), then $x_j$ is a point in $\mathbb{R}^N$. Define a Dirac measure as follows:

$$\delta_x(B) = \begin{cases} 1 & x \in B, \\ 0 & \text{other}. \end{cases} \tag{15}$$

It is easy to see that $\delta_x \in \mathcal{M}(\mathbb{R}^N)$. The Markov operator $M : \mathcal{M}(\mathbb{R}^N) \to \mathcal{M}(\mathbb{R}^N)$ of IFS, defined as

$$M\mu = \sum_{i=0}^{1} p_i \mu \circ f_i^{-1}, \mu \in \mathcal{M}(\mathbb{R}^N). \tag{16}$$

Now take $\mu_0 = \delta_{x_j}$, and the results of iterative calculation are as follows:

$$\mu_1 = M\mu_0 = \sum_{i=0}^{1} p_i \mu_0 \circ f_i^{-1} = \sum_{i=0}^{1} p_i \delta_{x_j} \circ f_i^{-1} = \sum_{i=0}^{1} p_i \delta_{f_i(x_j)} = \sum_{|\mathbf{i}|=1} \boldsymbol{p_i} \delta_{f_\mathbf{i}(x_j)}.$$

$$\begin{aligned}
\mu_2 &= M^2\mu_0 = M\mu_1 = \sum_{i=0}^{1} p_i \mu_1 \circ f_i^{-1} = \sum_{i=0}^{1} p_i (\sum_{|\mathbf{i}|=1} \boldsymbol{p_i} \delta_{f_\mathbf{i}(x_j)}) \circ f_i^{-1} \\
&= p_0 p_0 \delta_{f_0(f_0(x_j))} + p_0 p_1 \delta_{f_0(f_1(x_j))} + p_1 p_0 \delta_{f_1(f_0(x_j))} + p_1 p_1 \delta_{f_1(f_1(x_j))} \\
&= \sum_{|\mathbf{i}|=2} \boldsymbol{p_i} \delta_{f_\mathbf{i}(x_j)}
\end{aligned}$$

Inductively, we have

$$\mu_n = M^n \mu_0 = \sum_{|\mathbf{i}|=n} \boldsymbol{p_i} \delta_{f_\mathbf{i}(x_j)}. \tag{17}$$

By Theorem D.4, it follows from the Banach fixed point theorem that there exits a unique probability measure $\mu_*$ such that

$$\mu_n \xrightarrow{d_M} \mu_*.$$

The above $\mu_*$ is actually the self-similar measure of IFS. By Theorem D.5, we have $\mu_n \xrightarrow{w} \mu_*$, i.e.

$$\lim_{n\to\infty} \int F d\mu_n = \int F d\mu_*, \ \forall F \in C(\mathbb{R}^N).$$

It follows from (17) and (3) that

$$\begin{aligned}
\int F d\mu_* &= \lim_{n\to\infty} \int F d\mu_n = \lim_{n\to\infty} \int F d(\sum_{|\mathbf{i}|=n} \boldsymbol{p_i} \delta_{f_\mathbf{i}(x_j)}) \\
&= \lim_{n\to\infty} \sum_{|\mathbf{i}|=n} \boldsymbol{p_i} F(f_\mathbf{i}(x_j)) \\
&= \lim_{n\to\infty} \sum_{|\mathbf{i}|=n} \boldsymbol{p_i} F(\boldsymbol{H_i}[:, j]), \ \forall F \in C(\mathbb{R}^N). \tag{18}
\end{aligned}$$

In (18), $\forall i \in \{1, 2, ..., N\}$, take the continuous function $F_i$ to satisfy

$$F_i(\boldsymbol{t}) = t_i, \ \forall \boldsymbol{t} = (t_1, t_2, ..., t_N)^\top \in \mathbb{R}^N.$$

It follows from (18) that $\forall i \in \{1, 2, ..., N\}$,

$$\begin{aligned}
\lim_{n\to\infty} \boldsymbol{E}_n[i, j] &= \lim_{n\to\infty} \sum_{|\mathbf{i}|=n} \boldsymbol{p_i} \boldsymbol{H_i}[i, j] = \lim_{n\to\infty} \sum_{|\mathbf{i}|=n} \boldsymbol{p_i} F_i(\boldsymbol{H_i}[:, j]) \\
&= \int F_i(\boldsymbol{t}) d\mu_*(\boldsymbol{t}) = \int t_i d\mu_*(\boldsymbol{t}) = \int_{\mathbb{F}} t_i d\mu_*(\boldsymbol{t}) = e_i. \tag{19}
\end{aligned}$$

Which combined with simple mathematical analysis technology completes the proof.

## G  How to Set the Initial Value of Adjoint Probability Vector?

The geometric meaning of matrix determinant $\det \boldsymbol{A}$ is the expansion factor of graph volume under linear transformation $\boldsymbol{A}$. Let $\mathbb{F}$ be the fractal set of IFS as defined in IGNNS. Then

$$\mathbb{F} = f_0(\mathbb{F}) \bigcup f_1(\mathbb{F}).$$

Note that

$$\det \boldsymbol{A}_i = \frac{\text{volume}(f_i(\mathbb{F}))}{\text{volume}(\mathbb{F})}, \quad i = 0, 1.$$

It can be seen that if the value of $\det \boldsymbol{A}_i$ is large, it reflects that $f_i(\mathbb{F})$ has a large share in $\mathbb{F}$. therefore, when selecting the iterative function, $f_i$ should have a greater probability of being selected. So set

$$p_0 = \frac{\det \boldsymbol{A}_0}{\det \boldsymbol{A}_0 + \det \boldsymbol{A}_1} \quad, \quad p_1 = \frac{\det \boldsymbol{A}_1}{\det \boldsymbol{A}_0 + \det \boldsymbol{A}_1};$$

Note that $\boldsymbol{A}_0$, $\boldsymbol{A}_1$ are triangular matrixes and the diagonals of $triu(\boldsymbol{A})$, $tril(\boldsymbol{A})$ are equal to 1. It follows from $\boldsymbol{A}_0 = \boldsymbol{D}_0^{-1} tril(\boldsymbol{A})$ (or $\boldsymbol{A}_0 = \boldsymbol{D}_0^{-\frac{1}{2}} tril(\boldsymbol{A}) \boldsymbol{D}_0^{-\frac{1}{2}}$) that $\det \boldsymbol{A}_0 = \frac{1}{\det \boldsymbol{D}_0}$. Similarly, $\det \boldsymbol{A}_1 = \frac{1}{\det \boldsymbol{D}_1}$. It follows that

$$p_0 = \frac{\det \boldsymbol{D}_1}{\det \boldsymbol{D}_0 + \det \boldsymbol{D}_1} \quad, \quad p_1 = \frac{\det \boldsymbol{D}_0}{\det \boldsymbol{D}_0 + \det \boldsymbol{D}_1},$$

where $\boldsymbol{D}_0$ and $\boldsymbol{D}_1$ are degree matrices of $triu(\boldsymbol{A})$ and $tril(\boldsymbol{A})$ respectively.

