# OpenReview forum: "Iterated graph neural network system"
_ICLR.cc/2021/Conference — Reject_

### Official Review · AnonReviewer3 · 2020-10-21
**The description of the proposal could be improved**

**Rating:** 5
**Confidence:** 3

**Review:**

The paper proposes a new definition of GNNs designed to cope with bi-directional message-passing processes.  To do so, a new symbols space, different from the one adopted by Bidirectional GCN,  is considered, together with an iterated function system. These lead to an architecture composed of 4 steps: an input layer that acts as a classic FC layer; an IFS layer that applies the iterated function system considering the adjacency matrix; a layer to concatenate or sum the expected values of each iteration; and an output layer that combines the results using the functions of the IFS and a new learnable weight matrix.
Experiments on citations datasets show significantly better results than those obtained by many different related works.

The paper proposes an interesting approach, but personally, I found the document a bit foggy in some parts. On page 2, iterated function systems are introduced, but the definition does not explain what the f_i functions are, how many functions there may be and how the probabilities are related to the functions. This detail can be assumed in section 3.2 when the expected values are computed using the probabilities from the p set. However, in this section there are undefined symbols, such as p_{00}, p_{01}, etc., which make it difficult to follow the explanation of the step.

In section 3.2, a short discussion of the value of n (the number of iteration) should be added to say whether this value depends on the input graph, on the values of the H matrices in each iteration or something else.

In section 3.3, to compute the global representation R one can decide between two different definitions. These two definitions are very different and combine the results of each iteration. However, it is not clear to me how I should decide which one to use. I would like to see in the paper an explanation of what they represent and why it is more useful to calculate an average or a concatenation of the results of each iteration instead of considering only the results of the last iteration (which should depend on all the previous one).

As regards the experiments, the results seem promising, but information on trining time should be added.
Unfortunately, however,  the description of the datasets used is too simplified.  It should explain how the nodes are extracted from those available in the datasets and why only 20 nodes per class are extracted. It seems that only very few nodes are considered despite the availability of a large number of them. In addition, details about the edges connecting the nodes and the features can also be useful, as not all the edges/features will be considered since only 20 nodes per class are used. It is necessary to explain if these nodes belong to only one class or to several classes because there could be unbalance problems. Also, it is not specified which function is used in the representation layer.
It would be interesting to understand what happened for the Pubmed dataset. Probably the result depends on the small percentage of nodes considered.

Pros
- The results seem interesting

Cons
- Overall the proposal seems interesting, but its description lacks important definitions.
- Code and datasets are not available. Unfortunately, this does not help to evaluate the proposal.

Typos
The second line of section 3.4: labels instead of lebels
In theorem 4.2 furthermore instead of further more
In the references, arXiv URLs are not well-formed.

---

> ### Author Response · Authors · 2020-11-16
> **To Reviewer 3 (Part 1)**
>
> Thank you very much for your comments.
>
> __Comment1:__ The paper proposes an interesting approach, but personally, I found the document a bit foggy in some parts. On page 2, iterated function systems are introduced, but the definition does not explain what the f_i functions are, how many functions there may be and how the probabilities are related to the functions. This detail can be assumed in section 3.2 when the expected values are computed using the probabilities from the p set. However, in this section there are undefined symbols, such as p_{00}, p_{01}, etc., which make it difficult to follow the explanation of the step.
>
> __Reply__ Thank you very much for pointing out these mistakes. We have redescribed IFS in Chapter 2 and given a detailed definition of $f_i$. We have revised the error in formula (1). The correct case is that an $f_i$ corresponds to a probability $p_i$.  We forgot to thicken the font of P, resulting in undefined symbols. The correct definition is  __p__$_{01}$=$p_0p_1$.
>
>
> __Comment2:__  In section 3.2, a short discussion of the value of n (the number of iteration) should be added to say whether this value depends on the input graph, on the values of the H matrices in each iteration or something else.
>
> __Reply2:__  We have added "Let $n$ be the number of iterations of IFS. For IGNNS, $n$ is a preset parameter. " to section 3.2.  In section 3.6,  we also discuss the effect of $n$ on the time complexity of IGNNS.
>
> __Comment3:__  As regards the experiments, the results seem promising, but information on trining time should be added.
>
> __Reply3:__  In section 5.3, we have added training time information, including the total training time and the average training time per epoch. In section 3.6, We also discuss theoretical the time complexity of IGNNS.
>
> __Comment4:__  In section 3.3, to compute the global representation R one can decide between two different definitions. These two definitions are very different and combine the results of each iteration. However, it is not clear to me how I should decide which one to use.
>
> __Reply4:__  Indeed, this is a very useful question. So far, I can't prove theoretically which way is better. In my opinion, the most important consideration is the size of the parameter matrix $\mathbf{W}$ of the output layer. Suppose that the dimension of the hidden space is $H$, the dimension of the output layer is $P$, and the depth of the network is $n$. If the average operator is used, the parameter quantity of $\mathbf{W}$ is $HP$; if the concatenation operator is used, the parameter quantity of $\mathbf{W}$ is $nHP$. If $n$ and $H$ are both large, then the $nHP$ is also very large, which makes it difficult to train the network, especially for semi supervised experiments. If $H$ is less and $n$ is not large, we can consider concatenation operator. In short, I don't think there is a unified standard.

---

> ### Author Response · Authors · 2020-11-17
> **To Reviewer 3 (Part 2)**
>
> __Comment5:__  I would like to see in the paper an explanation of what they represent and why it is more useful to calculate an average or a concatenation of the results of each iteration instead of considering only the results of the last iteration (which should depend on all the previous one).
>
> Reply5:  We think that your problem is related to the core problem of deep GNN, that is, the overs-moothing problem.  We discussed this issue in Section 4. In short, the features of the last layer of deep GNN will become indistinguishable, so we often combine the previous features to overcome this smoothness.  In order to overcome the over-smoothing problem of deep GNN,  many new methods have been proposed, such as Skip connection[1], Drop edge[2], Residual connection [3], Identity mapping[4], Generalized message aggregation functions[5] and so on.
>
> Let's give an example. Let $\mathcal{G}$ be a completely connected graph with two nodes. Then the normalization adjacency matrix of $\mathcal{G}$ is $A=[[0.5,0.5],[0.5,0.5]]$.  Then for any $n$, $A^n=[[0.5,0.5],[0.5,0.5]]$. Let $X=[[x_1,y_1],[x_2,y_2]]$ be feature matrix of nodes. Then $A^nX=[[\frac{x_1+x_2}{2},\frac{y_1+y_2}{2}],[\frac{x_1+x_2}{2},\frac{y_1+y_2}{2}]]$. This is an ordinary feature.
>
> But for IGNNS, We will get an extraordinary feature (fractal representation).
> It is easy to see that $A_0=[[0.5,0.5],[0,1]]$ and  $A_1=[[1,0],[0.5,0.5]]$. Thus
>
> $\mathbb{H}^{(1)}$ ={$A_0X,A_1X$}={$[[\frac{x_1+x_2}{2},\frac{y_1+y_2}{2}],[x_2,y_2]]$, $[[x_1,y_1],[\frac{x_1+x_2}{2},\frac{y_1+y_2}{2}]]$}
> $\mathbb{H}^{(2)}$={$A_0A_0X,A_0A_1X$,$A_1A_0X,A_1A_1X$}
>
> At the same time, we also notice that the ergodic representation of IGNNS makes use of all the information obtained. But now the question is, is information all necessary? Is there redundancy? How to select the valid message from the fractal representation becomes the focus of our research in the next stage.
>
> [1]Keyulu Xu, Chengtao Li, Yonglong Tian, Tomohiro Sonobe, Ken ichi Kawarabayashi, and Stefanie
> Jegelka. Representation learning on graphs with jumping knowledge networks. In ICML, 2018.
> [2]Yu Rong, Wenbing Huang, Tingyang Xu, and Junzhou Huang. Dropedge: Towards deep graph
> convolutional networks on node classification. In International Conference on Learning Repre-
> sentations, 2020.
> [3]J. Klicpera, A. Bojchevski, and S. Gunnemann. Predict then propagate: Graph neural networks meet
> personalized pagerank. In ICLR, 2019a.
> [4]Ming Chen, Zhewei Wei, Zengfeng Huang, Bolin Ding, and Yaliang Li. Simple and deep graph
> convolutional networks. In International Conference on Machine Learning, 2020.
> [5]Guohao Li, Chenxin Xiong, Ali Thabet, and Bernard Ghanem. Deepergcn: All you need to train
> deeper gcns. 2020.

---

> ### Author Response · Authors · 2020-11-17
> **To Reviewer 3 (Part 3)**
>
> __Comment6:__ As regards the experiments, the results seem promising, but information on trining time should be added.
>
> __Reply6:__ We have added training time information in section 5.3, including the total training time and the average training time  per epoch. In section 3.6, we also analyze the time complexity of IGNNS.
>
> __Comment7:__ Unfortunately, however, the description of the datasets used is too simplified. It should explain how the nodes are extracted from those available in the datasets and why only 20 nodes per class are extracted. It seems that only very few nodes are considered despite the availability of a large number of them. In addition, details about the edges connecting the nodes and the features can also be useful, as not all the edges/features will be considered since only 20 nodes per class are used. It is necessary to explain if these nodes belong to only one class or to several classes because there could be unbalance problems.
>
> __Reply7:__ We have added a more detailed description of the dataset in Section 5.2，that is "In these citation
> networks, papers are represented as nodes, and citations of one paper by another are denoted as edges. Node features are the bag-of-words vector of papers, and node label is the only one academic topic of a paper."  So  these nodes belong to only one class.  For fair comparison,  we adopt the same data split mode as these baseline models, i.e. with 20 nodes per class for training, 500 nodes for validation and 1,000 nodes for testing. Because it is a semi-supervised node classification experiment, only a few nodes are needed for training. Indeed, it is interesting to make full use of the edge information of nodes in semi supervised experiments. In later research, we will try.
>
> __Comment8:__ Also, it is not specified which function is used in the representation layer.
>
> __Reply8:__  In Section 5.2, we have added the relevant description, that is “In representation layer, we adopt weighted time average to get the global representation of notes.”
>
> __Comment9:__  It would be interesting to understand what happened for the Pubmed dataset. Probably the result depends on the small percentage of nodes considered.
>
> __Reply9:__  Thank you very much for bringing this to our attention. We have analyzed this phenomenon in section 5.3. We already know  the improved performance of model IGNNS in dataset Cora and Citeseer is much higher than that in dataset Pubmed. To understand why this happens, we analyze the characteristics of these citation networks. We consider two statistical properties of networks, one is the Network Density $d(\mathcal{G})$, which is defined as $d(\mathcal{G})=\frac{2L}{N(N-1)}$, where $N$ is the number of nodes and $L$ is the number of edges, and the other is the Average Clustering Coefficient $C$, which is defined as $C=\frac{1}{N}\sum_{i\in V}C_i$, where $V$ is the set of nodes, $C_i=\frac{2e_i}{k_i(k_i-1)}$, $k_i$ is the number of the neighbors of node $v_i$ and $e_i$ is the number of undirected edges between $k_i$ neighbors. The less Network Density means the more global sparsity of the network, and the less Average Clustering Coefficient means the more sparsity of the neighbors of nodes. The calculation results of the statistical characteristics of the network are shown in Table 1 . We can see  that Pubmed is more sparse than Cora and Citeseer. The performance of IGNNS benefits from the bidirectional mixed propagation of information between nodes, but this sparsity weakens the gain of IGNNS.
>
> __Table1  Statistical characteristics of the networks. Bold for minimum.__
>
>  Statistical characteristics &emsp;&emsp;|&emsp; Cora &emsp;&emsp;| &emsp;&emsp;Citeseer &emsp; |&emsp;  Pubmed &emsp;
> :---- | :----: | :----: | :----:
> Network Density | 0.00144000 | 0.00084514 | __0.00022805__
> Average Clustering Coefficient| 0.24067330 | 0.14147102  |__0.06017521__
>
> __Comment10:__ Overall the proposal seems interesting, but its description lacks important definitions.
>
> __Reply10:__ We have corrected these errors.
>
> __Comment11:__ Code and datasets are not available. Unfortunately, this does not help to evaluate the proposal.
>
> __Reply11:__  We re submitted the code and made sure it works locally before committing.
>
> __Comment12:__ Typos The second line of section 3.4: labels instead of lebels In theorem 4.2 furthermore instead of further more In the references, arXiv URLs are not well-formed.
>
> __Reply12:__ Thank you very much for pointing out these mistakes and we have finished correcting them.

---

### Official Review · AnonReviewer1 · 2020-10-26
**An interesting method with insufficient discussion and evaluation**

**Rating:** 4
**Confidence:** 4

**Review:**

Overall, this paper proposes Iterated Graph Neural Network System, which provides a novel way for computing GNN messages.

However, there lack enough discussions with existing multi-layer GNNs.
The paper mentions that "the message passing in the two directions is independent and lacks of interaction". While this is true for a single layer GNN, when the GNN is multi-layer, the messages sent in deeper layers contains fused information from multiple directions.
Furthermore, if skip connections are used, the messages sent in deeper layers can have even richer information.
These discussions are lacking in the current paper.

Moreover, the evaluation is very insufficient.
Firstly, the paper mentions a General Framework in Section 5, including a new model R-IGNNS. However, no evaluation is made at all. I would regard the experiments as incomplete.
Additionally, there is no further analysis or ablation study provided. While the performance improvement seems to be hugel, without those analysis, it is really hard to understand where the improvement comes from.

More comments:
1 "Therefore, the above architectures can not deal with directed graph directly". I believe this is not the case: existing GNNs can naively work with directed graphs by doing message passing following the edge direction.

2 The paper mentions after Eq (3) that H^(n) has H x 2^n elements. Will it be a scalability concern?

---

> ### Author Response · Authors · 2020-11-16
> **To Reviewer 1 (Part 1)**
>
> Thank you very much for your comments.
>
> __Comment1:__ However, there lack enough discussions with existing multi-layer GNNs. The paper mentions that "the message passing in the two directions is independent and lacks of interaction". While this is true for a single layer GNN, when the GNN is multi-layer, the messages sent in deeper layers contains fused information from multiple directions. Furthermore, if skip connections are used, the messages sent in deeper layers can have even richer information. These discussions are lacking in the current paper.
>
> __Reply1:__ We define the direction as follows: first, we give the graph node a index number so that the $i$ row of the adjacency matrix represents the adjacency relation between node $i$ and other nodes. Thus, in this way, we can define the forward direction as $i\rightarrow j$ if $i<j$, where $i,j$ denote the index number of nodes. Similarly, we can define the backward direction as $i \leftarrow j$ if $i<j$.  For self adjacency node $i$, we can also define two directions (clockwise and anticlockwise).  In this sense, forward direction  matrix $\mathbf{A}_0$  is  the upper triangular matrix of $\mathbf{A}$ , backward direction matrix $\mathbf{A}_1$  is  the lower triangular matrix of $\mathbf{A}$.   In this sense, Bi-GCN (similar to Bi-LSTM) is equivalent to two independent GCNs induced by $\mathbf{A}_0$ and $\mathbf{A}_1$ respectively (simple matrix operation is needed to change $\mathbf{A}_i$ into symmetric matrix), only in output layer, two independent representations (generated by two independent GCNs) are concatenated together. Of course, you can define various directions. In order to avoid ambiguity, we have revised the relevant statements. In section 4 and section 5.4, We have added relevant discussion.
>
> __Comment2:__ Moreover, the evaluation is very insufficient. Firstly, the paper mentions a General Framework in Section 5, including a new model R-IGNNS. However, no evaluation is made at all. I would regard the experiments as incomplete.
>
> __Reply2:__  We initially included R-IGNNS in the paper in order to emphasize theoretically that IGNNS is a unified architecture. After re evaluation, we believe that R-IGNNS is an independent and complex work, which is worthy of further study and improvement. In addition, we believe that R-IGNNS does not affect the integrity of current work, so we delete the discussion on R-IGNNS in the newly revised papers.

---

> ### Author Response · Authors · 2020-11-16
> **To Reviewer 1 (Part 2)**
>
> __Comment3:__ Additionally, there is no further analysis or ablation study provided. While the performance improvement seems to be hugel, without those analysis, it is really hard to understand where the improvement comes from.
>
> __Reply3:__
>
> __1) We've added a new experiment, which is the performance of completely linear IGNNS (see section 5.4).__
>
> In Nonlinear IGNNS, we use the nonlinear activation function $\text{ReLU}(x)$, learn adjoint probability vector $\mathbf{p}=(p_0,p_1)$ by
> $$p_0\leftarrow\frac{\text{ReLU}(p_0)+0.1}{\text{ReLU}(p_0)+\text{ReLU}(p_1)+0.2}, p_1\leftarrow\frac{\text{ReLU}(p_1)+0.1}{\text{ReLU}(p_0)+\text{ReLU}(p_1)+0.2}$$ and learn the representation layer coefficient $\mathbf{r}=(r_1,r_2,...,r_n)$ by $r_i\leftarrow\text{ReLU}(r_i)$ with initial value $r_i=\left(\frac{1}{r}\right)^{i-1}$ where $r=\sqrt{\ln(N)+0.577215664}$.
> In this experiment, to get a completely linear IGNNS, we let all the activation functions be the identity function, i.e. $\sigma(x)=x$, and let the adjoint probability vector $\mathbf{p}=(p_0,p_1)$ and the representation layer coefficient $\mathbf{r}=(r_1,r_2,...,r_n)$ be hyperparameters without learning.
>
> __Table1   Performance of completely linear IGNNS__
>
>  Method &emsp;|&emsp; Cora &emsp; | &emsp; Citeseer &emsp;|&emsp;  Pubmed
> :---- | :----: | :----: | :----:
> GCN |  81.5| 70.3| 79.0
> IGNNS(Linear) |__83.9__| __72.4__ | __79.9__
>
> We can see from Table 1 that the performance of completely linear IGNNS is better than that of baseline model GCN. Compared with other models, completely linear IGNNS is still competitive. This is due to the fact that the IFS can extract more features than spectral filters. In the new version of the paper, more comments on this conclusion have been added.
>
> __2)  We analyze the original experimental results in detail (see section 5.3).__
>
> We already know  the improved performance of model IGNNS in dataset Cora and Citeseer is much higher than that in dataset Pubmed. To understand why this happens, we analyze the characteristics of these citation networks. We consider two statistical properties of networks, one is the Network Density $d(\mathcal{G})$, which is defined as $d(\mathcal{G})=\frac{2L}{N(N-1)}$, where $N$ is the number of nodes and $L$ is the number of edges, and the other is the Average Clustering Coefficient $C$, which is defined as $C=\frac{1}{N}\sum_{i\in V}C_i$, where $V$ is the set of nodes, $C_i=\frac{2e_i}{k_i(k_i-1)}$, $k_i$ is the number of the neighbors of node $v_i$ and $e_i$ is the number of undirected edges between $k_i$ neighbors. The less Network Density means the more global sparsity of the network, and the less Average Clustering Coefficient means the more sparsity of the neighbors of nodes. The calculation results of the statistical characteristics of the network are shown in Table 2 . We can see  that Pubmed is more sparse than Cora and Citeseer. The performance of IGNNS benefits from the bidirectional mixed propagation of information between nodes, but this sparsity weakens the gain of IGNNS.
>
> __Table2  Statistical characteristics of the networks. Bold for minimum.__
>
>  Statistical characteristics &emsp;&emsp;|&emsp; Cora &emsp;&emsp;| &emsp;&emsp;Citeseer &emsp; |&emsp;  Pubmed &emsp;
> :---- | :----: | :----: | :----:
> Network Density | 0.00144000 | 0.00084514 | __0.00022805__
> Average Clustering Coefficient| 0.24067330 | 0.14147102  |__0.06017521__
>
> __Comment4:__ "Therefore, the above architectures can not deal with directed graph directly". I believe this is not the case: existing GNNs can naively work with directed graphs by doing message passing following the edge direction.
>
> __Reply4:__  We think that the GNN based on spectral analysis cannot deal with directed graph directly because it requires the  adjacency matrix to be symmetric matrix. But there is no denying that it is possible to use some additional techniques to make it suitable for directed graphs. In order to avoid semantic fuzziness, we have modified the relevant statements.
>
> __Comment5:__ The paper mentions after Eq (3) that H^(n) has H x 2^n elements. Will it be a scalability concern?
>
> __Reply5:__ We want to emphasize that this is the maximum possible number. Which is the maximum amount of message that can be obtained by the IFS. When $n$ tends to infinity, it will produce fractal set, which is an uncountable set. So we call H^(n) the fractal representation of IGNNS. We also discuss this in section 5.4. In Appendix A, we present a visualization example, that is, Bi-GCN can only obtain the boundary of fractal set, while IGNNS can obtain all the information. At the same time, we also notice that the ergodic representation of IGNNS makes use of all the information obtained. But now the question is, is information all necessary? Is there redundancy? How to select the valid message from the fractal representation becomes the focus of our research in the next stage.

---

### Official Review · AnonReviewer4 · 2020-10-29
**A new graph neural networks architecture with modified rules for message passing**

**Rating:** 6
**Confidence:** 4

**Review:**

Summary:

This work proposes a new graph neural network architecture with modified rules for message passing, Iterated Graph Neural Network System (IGNNS). The paper then provides a theoretical analysis of the proposed architecture by connecting it with Iterated Function System (IFS), an important research field in fractal geometry. This paper further demonstrates empirically that the proposed architecture outperforms related models on citation network datasets.

Pros:

1.The proposed architecture achieves empirical improvement on citation network datasets.

2.This work also provides some theoretical analysis for the proposed architecture: it analyzes the geometric properties of IGNNS from the perspective of dynamical system.


Cons:
1.It seems unclear to me how the theoretical analysis via IFS could be used to explain the empirical performance gain on citation networks.

2.According to the formulas in equation (3) and above on page 4, it seems that the mathematical expectation $E_n$ is still linear (affine) w.r.t. to the input $X^{\text{int}}$. Then if we use a learnable matrix to learn such combinations of matrices $A_i$ and probability vector $p_i$, would this be equivalent to applying an MLP to for each the message passing iteration (the adjacency information in $A_i$  is accessible to update the MLP via backpropagation)?

3.In Figure 1, how would the message passing in d) be different compared to the message passing in c) after two iterations?

4.It would be nice for the authors to provide a theoretical analysis on the computational complexity for the proposed architecture IGNNS.

5.It would be nice for the authors to provide discussions with relevant works [1-3] on graph neural networks. [1] proposes a generalized aggregation function that allows successful and reliable training of very deep GCNs and how the proposed theorems in work could unify the mixed results (as Thm 4.1 and 4.2 state that the characterization ability of IGNNS would decrease with the increase of IFS iterations). [2] proposes a method for directional message passing as well. [3] proposes a theoretical framework for analyzing which type of GNN would achieve better generalization performance on the given task, which is also related to this paper for explaining the performance gain by IGNNS.

6.The quality of the writing could be much more improved. It would be nice for the authors to provide:
a) better intuitions on its analysis using IFS (e.g. what is the physical meaning of the fractal set in IFS and why is it important).
b) more connections between its proposed architecture and the empirical experiment section (e.g. how the proposed theorem could explain the performance gain connections)
There are also grammar mistakes in the paper which may hinder the understanding of the readers (e.g. last sentence in the abstract).

[1] Li, Guohao, et al. "Deepergcn: All you need to train deeper gcns." arXiv preprint arXiv:2006.07739 (2020).

[2] Klicpera, Johannes, Janek Groß, and Stephan Günnemann. "Directional message passing for molecular graphs." arXiv preprint arXiv:2003.03123 (2020).

[3] Xu, Keyulu, et al. "What Can Neural Networks Reason About?." arXiv preprint arXiv:1905.13211 (2019).

---

> ### Author Response · Authors · 2020-11-16
> **To Reviewer 4 (Part 1)**
>
> Thank you very much for your comments.
>
> __Comment1:__  It seems unclear to me how the theoretical analysis via IFS could be used to explain the empirical performance gain on citation networks.
>
> __Reply1:__  In response to your question, we present a visualization example (see Appendix A). For this idealized example,  Bi-GCN gets boundary messages of fractal set and IGNNS gets all messages of fractal set (Figure 3). In section 5.4, We also talked about the empirical performance gain on citation networks.
>
> Let's give a simple example to emphasize this point. Let $\mathcal{G}$ be a completely connected graph with two nodes. Then the normalization adjacency matrix of $\mathcal{G}$ is $A=[[0.5,0.5],[0.5,0.5]]$.  Then for any $n$, $A^n=[[0.5,0.5],[0.5,0.5]]$. Let $X=[[x_1,y_1],[x_2,y_2]]$ be feature matrix of nodes. Then $A^nX=[[\frac{x_1+x_2}{2},\frac{y_1+y_2}{2}],[\frac{x_1+x_2}{2},\frac{y_1+y_2}{2}]]$. This is an ordinary feature.
>
> But for IGNNS, We will get an extraordinary feature (fractal representation).
> It is easy to see that $A_0=[[0.5,0.5],[0,1]]$ and  $A_1=[[1,0],[0.5,0.5]]$. Thus
>
> $\mathbb{H}^{(1)}$ ={$A_0X,A_1X$}={$[[\frac{x_1+x_2}{2},\frac{y_1+y_2}{2}],[x_2,y_2]]$, $[[x_1,y_1],[\frac{x_1+x_2}{2},\frac{y_1+y_2}{2}]]$}
> $\mathbb{H}^{(2)}$={$A_0A_0X,A_0A_1X$,$A_1A_0X,A_1A_1X$}
>
> __Comments2:__ According to the formulas in equation (3) and above on page 4, it seems that the mathematical expectation $\mathbf{E}_n$ is still linear (affine) w.r.t. to the input $\mathbf{X}^{\text{int}}$. Then if we use a learnable matrix to learn such combinations of matrices$\mathbf{A}_i$  and probability vector $\mathbf{p}$, would this be equivalent to applying an MLP to for each the message passing iteration (the adjacency information in $\mathbf{A}_i$ is accessible to update the MLP via backpropagation)?
>
> __Reply2:__  Indeed, if $p_0, p_1$ are super parameters that does not need to be learned, then $\mathbf{E}_i$ is completely linear.   If $p_0, p_1$ are learnable， we learn them by
>
> $$p_0\leftarrow\frac{\text{ReLU}(p_0)+0.1}{\text{ReLU}(p_0)+\text{ReLU}(p_1)+0.2}, p_1\leftarrow\frac{\text{ReLU}(p_1)+0.1}{\text{ReLU}(p_0)+\text{ReLU}(p_1)+0.2}.$$
>
> We think this is an interesting question. Of course, MLP can approach $\mathbf{E}_i$  arbitrarily, however, the generalization ability still needs further study.  We do not use this method In IGNNS, because of the following considerations：
> (1) The introduction of learnable parameter matrix in IFS layer will not be the iterative result of IFS and can not be consistent in mathematical form;
> (2) Even if the shared parameter matrix is used in each layer, the parameter quantity is equal to $N^2$ ($N$ is the number of nodes), which is difficult to train for large graphs.
> (3) A positive number multiplied by the adjacency matrix $\mathbf{A}$ of graph does not change the inherent adjacency relation of nodes, which is what we need to get. Therefore, we only use  $\mathbf{p}_i\mathbf{A}_i$  to keep the multi-hoop or interactive relationship of graph nodes, where $i=i_1i_2...i_m$ .  However, $W\mathbf{A}_i$ may destroy the graph structure when you use it as input for the next iteration, where $W\in \mathbb{R}^{N\times N}$  a learnable parameter matrix.

---

> ### Author Response · Authors · 2020-11-16
> **To Reviewer 4 (Part 2)**
>
> __Comment3:__ In Figure 1, how would the message passing in d) be different compared to the message passing in c) after two iterations?
>
> __Reply3:__ Let's use an idealized example to illustrate the difference, which is  the lossy message passing model of gaph $\mathcal{G}$ with only one self adjacent node $v$. For the sake of discussion, we assume that the dimension of the hidden space is equal to 1. This means that the input of GNN (Bi-GCN or IGNNS) is a point in $\mathbb{R}$. For convenience, we ignore the activation function(regard as $\sigma(x)=x$) and parameter matrix (regard as $\mathbf{I}_{n,m}$). We assume that messages are sent from node $v$, propagate in two directions (clockwise and anticlockwise), and are finally received by node $v$. The received messages in the clockwise direction become one-third of the original, and the received messages in the anticlockwise direction become one-third of the original plus a constant of $\frac{2}{3}$. It is expressed by mathematical formula as follows
> $$f_0(x)=\frac{x}{3},\quad f_1(x)=\frac{x}{3}+\frac{2}{3},\quad x\in\mathbb{R}.$$
> For Bi-GCN, messages are delivered independently in both directions. In other words, there are two independent channels, and the message passing (transmitting or receiving) can only be carried out by their own channels. Let $x_0\in\mathbb{R}$ be the initial message. In the clockwise direction, after $n$ passes, the received message is
> $$\frac{x_0}{3}, \frac{x_0}{3^2},..., \frac{x_0}{3^n}\rightarrow0.$$
> In the anticlockwise direction, the received message is
> $$\frac{x_0}{3}+\frac{2}{3}, \frac{x_0}{3^2}+\frac{2}{3^2}+\frac{2}{3},..., \frac{x_0}{3^n}+\sum_{i=1}^n\frac{2}{3^i}\rightarrow1.$$
> For IGNNS, the two channels have a connection point at node $v$. first, node $v$ sends the message $x_0$ in both directions, and the connection point of node $v$ will receive two message $\{f_0(x_0), f_1(x_0)\}$. In the second launch, any message $(f_0(x_0)$ or $f_1(x_0))$ can be sent in both directions, so the received message is $$\{f_0(f_0(x_0)),f_0(f_1(x_0)),f_1(f_0(x_0)),f_1(f_1(x_0))\}.$$ In summary, After $n$ passes, the received message is $$\mathbb{H}^{(n)}=\{f_{\textbf{i}}(x_0)\}_{|\textbf{i}|=n}\rightarrow C,$$ where $C$ is the famous Cantor Set.
>
> __Comment4:__ It would be nice for the authors to provide a theoretical analysis on the computational complexity for the proposed architecture IGNNS.
>
> __Reply4:__ In the section 3.6 of the newly  revised papers， we give theoretical time complexity of IGNNS. That is
> $$\text{T}(\text{IGNNS})= O(2^{n}N^2H+N^2P)\quad \text{or} \quad O(2^{n}N^2H+N^2P+nNHP).$$
> Where $n$ is the number of iterations of IFS, $N$ is the number of nodes, $H$ is the dimension of the latent space and $P$ is the dimension of the output layer. In practice, for large graphs, $2^{n}N^2H\gg N^2P \gg nNHP$, thus $2^{n}N^2H$ is the main factor affecting the time complexity of IGNNS. Furthermore, for large graphs of the same size, $n$ is the main important factor affecting time complexity.

---

> ### Author Response · Authors · 2020-11-16
> **To Reviewer 4 (Part 3)**
>
> __Comment5:__  It would be nice for the authors to provide discussions with relevant works [1-3] on graph neural networks. [1] proposes a generalized aggregation function that allows successful and reliable training of very deep GCNs and how the proposed theorems in work could unify the mixed results (as Thm 4.1 and 4.2 state that the characterization ability of IGNNS would decrease with the increase of IFS iterations). [2] proposes a method for directional message passing as well. [3] proposes a theoretical framework for analyzing which type of GNN would achieve better generalization performance on the given task, which is also related to this paper for explaining the performance gain by IGNNS.
>
> __Reply5:__ Thank you very much for letting us know about these studies which are closely related to our work. We have discussed these works in the newly revised paper( section 1 and section 4 ).
>
> __Comment6:__ The quality of the writing could be much more improved. It would be nice for the authors to provide: a) better intuitions on its analysis using IFS (e.g. what is the physical meaning of the fractal set in IFS and why is it important). b) more connections between its proposed architecture and the empirical experiment section (e.g. how the proposed theorem could explain the performance gain connections) There are also grammar mistakes in the paper which may hinder the understanding of the readers (e.g. last sentence in the abstract).
>
> __Reply6:__
> 1）I added a new explanation to the original experiment results (section 5.3). We conclude that IGNNS is more suitable for dense networks, which is in line with our experience. The performance of IGNNS benefits from the bidirectional mixed propagation of information between nodes, but the sparsity of network will weakens the gain of IGNNS.
> 2)  We've added a new experiment, which is the performance of completely linear IGNNS (see section 5.4). The experimental results show that completely linear IGNNS is better than that of baseline model GCN. Compared with other models, completely linear IGNNS is still competitive. Combining with IFS, fractal set and over-smoothing problem of depth GNN, we fully discuss the experimental results.
> 3) We present a visualization example (see Appendix A).  This example vividly shows that Bi-GCN and IGGNS are connected by fractal set generated by iterated function system, i.e. Bi-GCN gets boundary messages and IGNNS gets all messages.
> 4) We have read through the paper many times and tried to correct all the mistakes.

---

### Official Review · AnonReviewer2 · 2020-10-31
**Review for Iterated graph neural network system**

**Rating:** 6
**Confidence:** 2

**Review:**

This paper proposes a new framework of GNN which can deal with undirected and directed graphs in a unified way. The authors argue that the size of the symbol space for a message passing path with length n is 2^n, while previous architectures only have constant size. Motivated by this observation, the authors borrow ideas from Iterated Function System to augment the symbol space.

Roughly speaking, the main idea is to use 2 linear mappings f_0 and f_1 which correspond to the two directions. For the given input vector x, in each of the n iterations, we apply one of the two linear mappings randomly (the probability of applying each mapping is a learnable parameter), and we use the expectation of the resulting vector as the representation. The authors argue that the resulting symbol space could have sufficient size if n is sufficiently large.

The main idea looks reasonable and interesting. My major concern is about the experiment part (and thus my recommendation would just be weak acceptance) . The authors only perform experiments on three datasets, and it is unclear if the same approach will be effective on other datasets/settings. Moreover, although we see performance improvement on two of the three datasets, necessary discussion about the experimental results is missing. It would be great if the authors could explain why the proposed method improves the performance on Cora and Citeseer, and achieves worse performance compared to some of the baselines on Pubmed, to give the readers a better idea when this new framework would be effective.

***Post Rebuttal***
I have read authors' response and other reviewers' reviews. After reading it is still unclear to me why sparsity of the networks could affect the performance of the proposed framework. Therefore I would like to keep my original score.

---

> ### Author Response · Authors · 2020-11-16
> **To Reviewer 2.  We added new experiments and analyzed all experimental results.**
>
> Thank you very much for your comments.
>
> __Comments:__  My major concern is about the experiment part (and thus my recommendation would just be weak acceptance) . The authors only perform experiments on three datasets, and it is unclear if the same approach will be effective on other datasets/settings. Moreover, although we see performance improvement on two of the three datasets, necessary discussion about the experimental results is missing. It would be great if the authors could explain why the proposed method improves the performance on Cora and Citeseer, and achieves worse performance compared to some of the baselines on Pubmed, to give the readers a better idea when this new framework would be effective.
>
> __Reply:__
>
>
> #1) We've added a new experiment, which is the performance of completely linear IGNNS (see section 5.4).
>
> In Nonlinear IGNNS, we use the nonlinear activation function $\text{ReLU}(x)$, learn adjoint probability vector $\mathbf{p}=(p_0,p_1)$ by
> $$p_0\leftarrow\frac{\text{ReLU}(p_0)+0.1}{\text{ReLU}(p_0)+\text{ReLU}(p_1)+0.2}, p_1\leftarrow\frac{\text{ReLU}(p_1)+0.1}{\text{ReLU}(p_0)+\text{ReLU}(p_1)+0.2}$$ and learn the representation layer coefficient $\mathbf{r}=(r_1,r_2,...,r_n)$ by $r_i\leftarrow\text{ReLU}(r_i)$ with initial value $r_i=\left(\frac{1}{r}\right)^{i-1}$ where $r=\sqrt{\ln(N)+0.577215664}$.
> In this experiment, to get a completely linear IGNNS, we let all the activation functions be the identity function, i.e. $\sigma(x)=x$, and let the adjoint probability vector $\mathbf{p}=(p_0,p_1)$ and the representation layer coefficient $\mathbf{r}=(r_1,r_2,...,r_n)$ be hyperparameters without learning.
>
> __Table1   Performance of completely linear IGNNS__
>
>  Method &emsp;|&emsp; Cora &emsp; | &emsp; Citeseer &emsp;|&emsp;  Pubmed
> :---- | :----: | :----: | :----:
> GCN |  81.5| 70.3| 79.0
> IGNNS(Linear) |__83.9__| __72.4__ | __79.9__
>
> We can see from Table 1 that the performance of completely linear IGNNS is better than that of baseline model GCN. Compared with other models, completely linear IGNNS is still competitive. This is due to the fact that the IFS can extract more features than spectral filters. In the new version of the paper, more comments on this conclusion have been added.
>
> #2)  We analyze the original experimental results in detail (see section 5.3).
>
> We already know  the improved performance of model IGNNS in dataset Cora and Citeseer is much higher than that in dataset Pubmed. To understand why this happens, we analyze the characteristics of these citation networks. We consider two statistical properties of networks, one is the Network Density $d(\mathcal{G})$, which is defined as $d(\mathcal{G})=\frac{2L}{N(N-1)}$, where $N$ is the number of nodes and $L$ is the number of edges, and the other is the Average Clustering Coefficient $C$, which is defined as $C=\frac{1}{N}\sum_{i\in V}C_i$, where $V$ is the set of nodes, $C_i=\frac{2e_i}{k_i(k_i-1)}$, $k_i$ is the number of the neighbors of node $v_i$ and $e_i$ is the number of undirected edges between $k_i$ neighbors. The less Network Density means the more global sparsity of the network, and the less Average Clustering Coefficient means the more sparsity of the neighbors of nodes. The calculation results of the statistical characteristics of the network are shown in Table 2 . We can see  that Pubmed is more sparse than Cora and Citeseer. The performance of IGNNS benefits from the bidirectional mixed propagation of information between nodes, but this sparsity weakens the gain of IGNNS.
>
> __Table2  Statistical characteristics of the networks. Bold for minimum.__
>
>  Statistical characteristics &emsp;&emsp;|&emsp; Cora &emsp;&emsp;| &emsp;&emsp;Citeseer &emsp; |&emsp;  Pubmed &emsp;
> :---- | :----: | :----: | :----:
> Network Density | 0.00144000 | 0.00084514 | __0.00022805__
> Average Clustering Coefficient| 0.24067330 | 0.14147102  |__0.06017521__

---

### Decision · Program_Chairs · 2021-01-07
**Final Decision**

**Decision:**

Reject

**Comment:**

Reviewers found the new framework interesting. However, reviewers are unsatisfied with empirical evaluations. More experiments and discussion are needed.